# Motion of VAPB molecules reveals ER–mitochondria contact site subdomains

Christopher J. Obara[1,12 ✉], Jonathon Nixon-Abell[1,2,6,12], Andrew S. Moore[1], Federica Riccio[1,2,7], David P. Hoffman[1,8], Gleb Shtengel[1], C. Shan Xu[1,9], Kathy Schaefer[1], H. Amalia Pasolli[1], Jean-Baptiste Masson[3], Harald F. Hess[1], Christopher P. Calderon[4,5], Craig Blackstone[2,10,11] & Jennifer Lippincott-Schwartz[1 ✉]

To coordinate cellular physiology, eukaryotic cells rely on the rapid exchange of molecules at specialized organelle–organelle contact sites[1,2]. Endoplasmic reticulum–mitochondrial contact sites (ERMCSs) are particularly vital communication hubs, playing key roles in the exchange of signalling molecules, lipids and metabolites[3,4]. ERMCSs are maintained by interactions between complementary tethering molecules on the surface of each organelle[5,6]. However, due to the extreme sensitivity of these membrane interfaces to experimental perturbation[7,8], a clear understanding of their nanoscale organization and regulation is still lacking. Here we combine three-dimensional electron microscopy with high-speed molecular tracking of a model organelle tether, Vesicle-associated membrane protein (VAMP)-associated protein B (VAPB), to map the structure and diffusion landscape of ERMCSs. We uncovered dynamic subdomains within VAPB contact sites that correlate with ER membrane curvature and undergo rapid remodelling. We show that VAPB molecules enter and leave ERMCSs within seconds, despite the contact site itself remaining stable over much longer time scales. This metastability allows ERMCSs to remodel with changes in the physiological environment to accommodate metabolic needs of the cell. An amyotrophic lateral sclerosis-associated mutation in VAPB perturbs these subdomains, likely impairing their remodelling capacity and resulting in impaired interorganelle communication. These results establish high-speed single-molecule imaging as a new tool for mapping the structure of contact site interfaces and reveal that the diffusion landscape of VAPB at contact sites is a crucial component of ERMCS homeostasis.

The most prevalent sites of contact between organelles in mammalian cells are between the endoplasmic reticulum (ER) and mitochondria[9,10]. This interface has been implicated in a multitude of biological processes in both health and disease, ranging from lipid synthesis and catabolism to calcium signalling and facilitation of cellular respiration[2,3,11]. Commonly referred to as ER–mitochondria contact sites (ERMCSs) or mitochondria–ER contact sites (MERCs), these structures are ubiquitous in eukaryotes and may exist in several distinct forms (see Supplementary Text, sections 2a–b for full discussion)[3,4,6,12]. Nevertheless, technical limitations have limited our understanding of many aspects of ERMCS biology, as the structures are exquisitely sensitive to fixation artefacts and show remarkable biological heterogeneity even in single cells. As a result, biochemistry-based approaches often suffer from averaging artefacts, and most existing labelling technologies are perturbative to

ERMCS size, structure and regulation (see Supplementary Text, sections 1 and 2 for discussion)[7]. In particular, our understanding of the dynamic regulation of their molecular components and substructure is lacking. Here, we introduce quantitative imaging approaches that combine three-dimensional (3D) electron microscopy and live cell, high-speed single-molecule imaging to describe the substructure and dynamics of molecules within ERMCSs under different physiological and pathological conditions.

Specific sites of contact between ER and mitochondria are mediated in trans by pairs of molecular tethers (Fig. 1a)[1,5,6]. One well-established ERMCS tether is the ER-localized Vesicle-associated membrane protein (VAMP)-associated protein B (VAPB)[13–15], which interacts with mitochondrial binding partners to facilitate calcium and lipid transfer[16,17] and can harbour mutations that cause a severe form of motor neuron

[1]Janelia Research Campus, Howard Hughes Medical Institute, Ashburn, VA, USA. [2]Neurogenetics Branch, National Institute of Neurological Disorders and Stroke, NIH, Bethesda, MD, USA. [3]Decision and Bayesian Computation, Neuroscience, & Computational Biology Departments, CNRS UMR 3751, Institut Pasteur, Université de Paris, Paris, France. [4]Department of Chemical and Biological Engineering, University of Colorado Boulder, Boulder, CO, USA. [5]Ursa Analytics, Inc., Denver, CO, USA. [6]Present address: Cambridge Institute for Medical Research (CIMR), Cambridge, UK. [7]Present address: Centre for Gene Therapy & Regenerative Medicine, King's College London, London, UK. [8]Present address: 10x Genomics, Pleasanton, CA, USA. [9]Present address: Department of Cellular and Molecular Physiology, Yale University School of Medicine, New Haven, CT, USA. [10]Present address: MassGeneral Institute for Neurodegenerative Disease, Massachusetts General Hospital, Charlestown, MA, USA. [11]Present address: Department of Neurology, Massachusetts General Hospital and Harvard Medical School, Boston, MA, USA. [12]These authors contributed equally: Christopher J. Obara, Jonathon Nixon-Abell. ✉e-mail: obarac@janelia.hhmi.org; lippincottschwartzj@janelia.hhmi.org

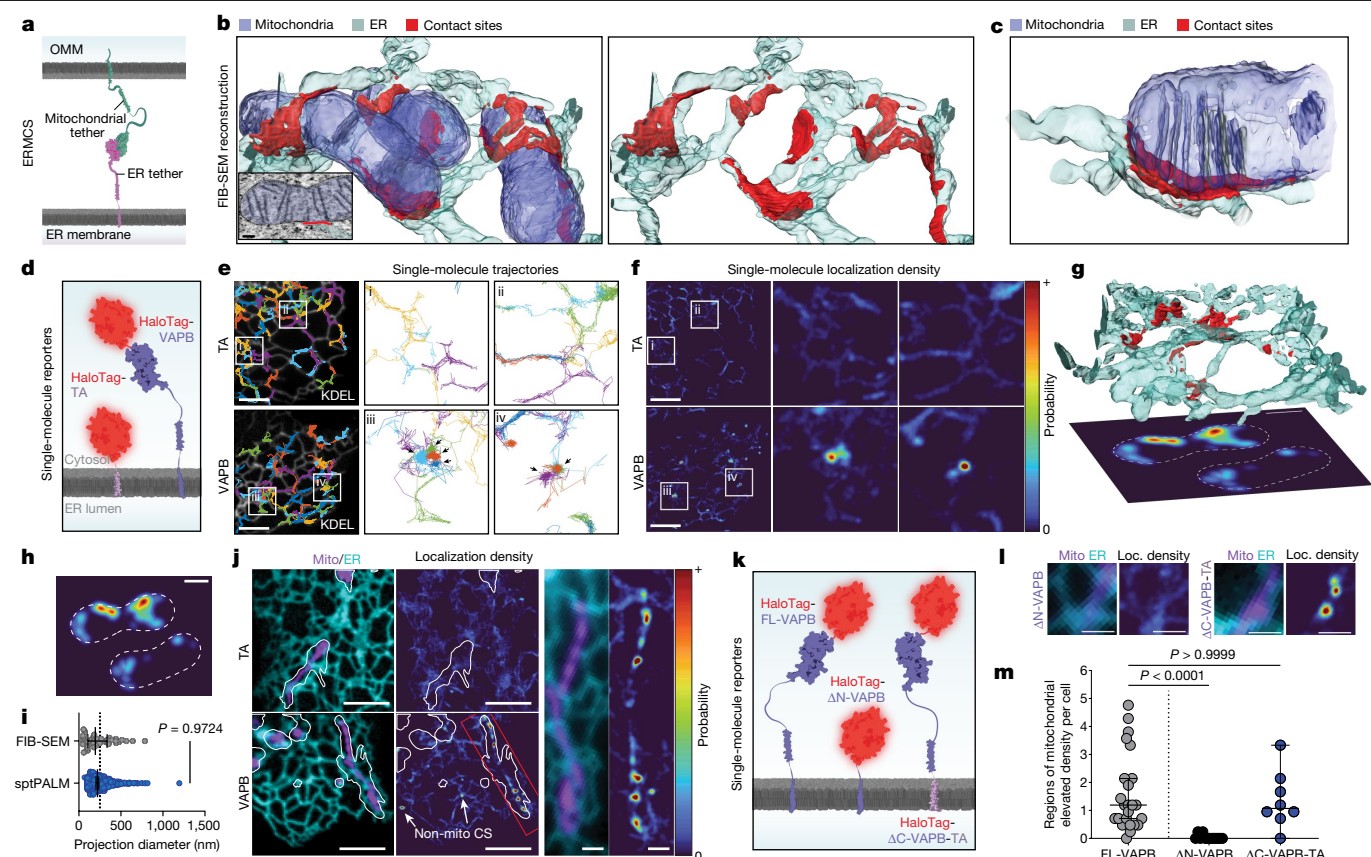

**Fig. 1 | Altered ER tether motion at ER–mitochondria contact sites.**
**a**, Cartoon indicating ER–mitochondrial tethering. **b**, 3D FIB-SEM reconstruction of ER (cyan), mitochondria (blue) and ERMCSs (red). Inset shows representative EM slice overlaid with segmentation masks. **c**, 3D reconstruction of an ERMCS with aligned cristae. **d**, Single-molecule tracers used in **e**–**f**. **e**, sptPALM trajectories of either non-specific tail anchor (TA, top) or VAPB (bottom) overlaid on ER reference image (KDEL, white). Arrows indicate regions of spatially correlated motion. **f**, Spatial probability maps for TA (top) or VAPB (bottom) generated from trajectories in **e**. Hotspots indicate likely sites of VAPB tethering. **e**–**f** are representative examples of n = 13 (TA, regions i and ii) or n = 24 (VAPB, regions iii and iv) cells. **g**, 3D EM reconstruction of contact sites (CS) (red) between ER (cyan) and mitochondria (not shown), with simulated sptPALM localization densities from the volume projected below. Dashed lines indicate mitochondria borders. **h**, Simulated localization densities from **g**. **i**, Contact size measured from FIB-SEM simulation projections or from single-molecule localization density (n = 466 contact sites (sptPALM), 38 contact sites (FIB-SEM); P = 0.9724, Dunn's multiple comparisons test, two-sided). Dotted line indicates confocal microscopy resolution limit. **j**, Micrograph of ensemble ER and mitochondria reporters (left), with corresponding TA or VAPB probability maps (centre) and magnified inset (right) in an example cell. Outlines indicate the area explored by each mitochondrion during the experiment. See Extended Data Fig. 2 for full dataset. **k**, Cartoon indicating VAPB single-molecule tracers used in **l**–**m**. ΔN-VAPB, deletion of the N terminus; ΔC-VAPB-TA; replacement of the C terminus with TA. **l**, Representative ERMCSs with associated localization (Loc.) densities for the tracer molecules shown in **k**. **m**, Number of mitochondria-associated regions of enhanced tethering per cell for the indicated VAPB reporters (n = 24 (VAPB), 12 (ΔN-VAPB), 8 (ΔC-VAPB-TA); P < 0.0001, P > 0.9999, Dunn's multiple comparisons test, two-sided). Error bars show the median and 95% confidence interval. Scale bars, 100 nm (**b**), 5 μm (**e**,**f**,**j**), 1 μm (**j**, inset), 500 nm (**g**–**h**), 1 μm (**l**). All panels are representative of at least two independently performed experiments with similar results.

disease[18,19]. The best characterized binding partner for VAPB in the mitochondria is the outer mitochondrial membrane (OMM) component, protein tyrosine phosphatase interacting protein 51 (PTPIP51)[14,16,20]. To visualize the 3D architecture of ERMCSs where VAPB and PTPIP51 could potentially interact, we used high pressure freezing followed by freeze-substitution and focused ion beam-scanning electron microscopy (FIB-SEM)[21]. This approach preserves contact sites in their near native state[8], avoiding chemical fixation artefacts that might disrupt their delicate organization[22–24] (Supplementary Text, section 3a). ER and mitochondria in small volumes from several COS7 cells were manually reconstructed. We identified contact sites as regions of ER membrane within 24 nm of the OMM, the predicted tethering distance of VAPB[12,20] (Fig. 1b, Extended Data Fig. 1a–c and Supplementary Video 1; see Methods and Supplementary Text, section 3b, for details). ERMCSs were highly abundant in the FIB-SEM volumes, often appearing near the base of mitochondrial cristae (Fig. 1c and Extended Data Fig. 1c) or at regions of constriction in the OMM (Extended Data Fig. 1b). The

placement of these sites is consistent with the known roles of ERMCSs in mitochondrial energy production and mitochondrial fission[25–27].

We next sought to characterize ERMCSs at the molecular level in the dynamic setting of a living cell. Single particle tracking-photoactivation localization microscopy (sptPALM)[28] was used to follow the motion of individual VAPB molecules, while simultaneously capturing the location of the ER, a necessary requirement to minimize artefacts in forming trajectories (Supplementary Video 2 and Supplementary Text, sections 6a–b). Single molecules of VAPB and other proteins in this system were visualized by genetically fusing them to a HaloTag and labelling them with a photoactivatable version of JF646 (ref. 29), which was photoconverted at very low efficiency to ensure single proteins were tracked correctly with minimal linking artefacts (Fig. 1d and Supplementary Video 2).

To interpret patterns of molecular motion, we first used a probe that does not associate with contact sites, a HaloTag-targeted to the ER with a minimal tail anchor (HaloTag-TA). We found that it explored

the surface of the ER randomly, showing no detectable preferences for any specific ER regions (Fig. 1e, tail anchor). By contrast, when we then tracked VAPB, we observed clusters of highly spatially associated trajectories (Fig. 1e, VAPB, see black arrows). By converting all trajectories within a specific time window to a spatially defined probability function (Methods), locations of spatially associated trajectories appeared as ER regions with elevated probability of VAPB localization (Fig. 1f). These probability "hotspots" were present in the VAPB dataset but not with Halo-TA (Fig. 1f), and they were consistent with the size of ERMCSs extracted from FIB-SEM volumes when analysed at a similar spatial resolution (Fig. 1g–i and Extended Data Fig. 1d, see Methods for more details).

Simultaneous imaging of ER and mitochondria in conjunction with sptPALM confirmed that the majority of VAPB hotspots were located on regions of ER close to mitochondria (Fig. 1j and Supplementary Video 2), consistent with hotspots being areas of increased ER–mitochondria tethering. Of note, some probability hotspots also occurred in regions of ER with no mitochondria nearby (Fig. 1j, arrows, and Extended Data Fig. 2), as expected due to VAPB's known role as a tether for additional organelles[14,15]. The formation of VAPB hotspots was entirely dependent on the ability of the protein to act as a tether by binding its interaction partner through its N-terminal cytosolic domain. Truncation of this N-terminal domain abrogated hotspot formation and, conversely, fusion of the N-terminal domain to the ER-targeted TA control was sufficient to form hotspots (Fig. 1k–m). We thus concluded that VAPB-associated hotspots at mitochondria represented tether interactions at bona fide ERMCSs.

In addition to providing a super-resolved image of ERMCSs in living cells, our sptPALM-based approach permitted analysis of how individual VAPB molecules interacted with an ERMCS over time. Notably, not all VAPB molecules that encountered a contact site showed direct engagement, and the probability of engagement was variable between ERMCSs, even in a single cell (Extended Data Fig. 3a–e). Remarkably, engaged VAPB molecules resided within a contact site only very briefly, with most molecules leaving the site on millisecond time scales (Fig. 2a–d and Supplementary Video 3, median dwell time = 556 ms). This dynamic exchange was nearly ten times faster than that reported for molecules at sites of ER contact with the plasma membrane (Supplementary Text, section 2c–d), underscoring the rapid rearrangements of VAPB and other molecules at this interface.

To understand these transient interactions more quantitatively, we utilized a non-parametric Bayesian approach[30]. Trajectories of sufficient length (greater than 500 steps, approximately 5.5 s) were broken into segments that were estimated to represent distinct kinetic states[31] (Supplementary Text, sections 6d–e). This allowed classification of segments that were either freely diffusing in the ER (blue) or contact site-associated (red) (Fig. 2e,f and Extended Data Fig. 4a–c). When analysed over many contact sites in multiple cells, we found that single VAPB molecules in ERMCSs showed significantly reduced diffusion ($D_{eff}$) relative to VAPB molecules in surrounding ER tubules (Fig. 2g). At the same time, VAPB molecules within ERMCSs retained signatures of diffusive motion compared to immobilized bead controls (Fig. 2h; see Supplementary Text, section 6d–e, for details), indicating that VAPB molecules within contact sites still underwent dynamic motion (see Supplementary Text, section 2c, for discussion).

Although some ERMCSs moved during the experiment or were short-lived (less than 1 min) (Extended Data Fig. 5), most remained stable over 60–90 s of imaging. During this time, many individual VAPB molecules repeatedly explored the same contact site interface. To map the diffusion landscape at this interface, we divided the contact site into small regions and analysed the average diffusion of trajectory segments within each neighbourhood (Extended Data Fig. 6a–f and Supplementary Text, section 6c)[32,33]. Examining the diffusive behaviour of VAPB in each stable contact site in our dataset, we discovered a consistent pattern in the VAPB diffusion landscape. Every contact site examined shared a central subdomain of low diffusion that gradually returned to the diffusion landscape of the surrounding ER at the edges of the contact site (Fig. 2i–k). The location and shape of this low diffusion subdomain closely correlated to the likelihood of finding a VAPB molecule (Fig. 2k), indicating that sites of slowed diffusion in the central subdomain were also elevated in tether abundance. This suggested that the VAPB-enriched central subdomain, where VAPB tether concentration was greatest, could represent the region of the contact site where adhesive forces were greatest.

To determine whether there were areas of increased adhesion between ER and mitochondria, and if so, what their distribution was across the contact site, we examined ERMCSs in our FIB-SEM data sets. Examining local curvature of the ER in each contact site (Supplementary Text, section 3d–e), we found that the ER membrane had a distinct region of net negative curvature at its centre (Fig. 2l,m, arrows). In this core region of the ERMCS, the ER adopted the inverse shape of the mitochondrion to which it was bound. This suggested adhesive tethering in this region is sufficient to deform the ER membrane from its normal high degree of positive mean curvature observed outside of the contact site (Extended Data Fig. 7a–f).

To address whether these regions of negative local ER curvature correlated with the VAPB low diffusion subdomains described above, we aligned all sptPALM-detected contact sites localized along simple tubular ER structures to the direction of the incoming ER tubule (as done in Extended Data Fig. 7 for FIB-SEM). The summed probability density at these contact sites afforded increased resolution and showed that the distinct peak of VAPB intensity had clear probability shoulders extending into the surrounding ER tubule (Fig. 2n, red and grey bars, and Extended Data Fig. 6g). This resulting probability distribution closely resembled the pattern of negative/positive curvature areas within contact sites from FIB-SEM data sets (Fig. 2m). Therefore, both structural FIB-SEM data and patterns of VAPB molecular diffusion support the presence of a central subdomain within ERMCSs that has higher VAPB abundance and greater adhesion properties. The gradual change in VAPB diffusion and tether abundance moving into the central subdomain suggest contact sites may exist as metastable interfaces. In this model, ERMCSs would be characterized by many transient binding events, whose likelihood gradually increases toward the central subdomain, consistent with the rapid tether exchange observed above (Fig. 2a–d). Such a model would predict adhesive tethering forces in the contact site are likely not evenly spread throughout the structure, instead being preferentially enriched at the central subdomain, resulting in the observed negative curvature.

To understand how the overall size of ERMCSs is regulated, we examined ERMCS size dependency on the availability of an associated tethering partner. We transiently overexpressed PTPIP51 (Fig. 3a) in COS7 cells and performed sptPALM experiments. Contact site size significantly increased and covered large portions of the mitochondria compared to control cells not expressing PTPIP51 (Fig. 3b). The ERMCS expansion occurred across both axes of the structure (Fig. 3c,d). To test whether contact site expansion was equally dependent on both members of the tethering pair, we labelled the ER and mitochondria in COS7 cells and overexpressed either VAPB or PITPIP51, evaluating ER and mitochondrial morphology by Airyscan imaging. ER and mitochondrial morphology were insensitive to overexpression of VAPB at the time points assayed, but there were dramatic rearrangements upon overexpression of PTPIP51, with the ER now completely enveloping most mitochondria (Fig. 3e). Despite this dramatic structural rearrangement, analysis of the sptPALM data showed no increase in the total number of contact sites per cell. There was, however, a loss of nearly all VAPB contact sites associated with organelles other than mitochondria and a significant enrichment in the proportion of VAPB on mitochondria at any moment (Fig. 3f). This suggested that in this condition VAPB-binding at the mitochondria outcompeted its potential interactions at non-mitochondria associated sites. In agreement,

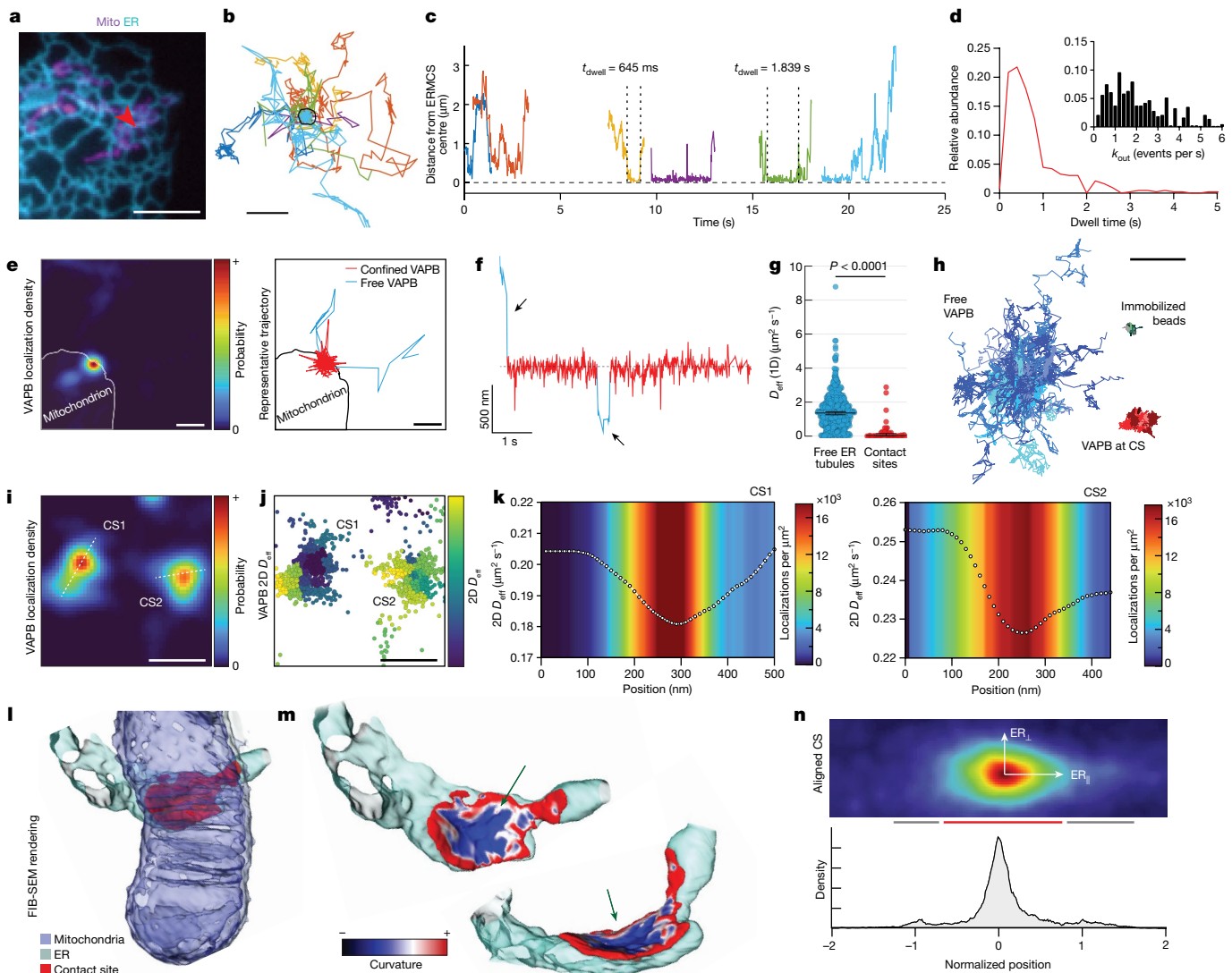

**Fig. 2 | Dynamic interactions generate a variable VAPB diffusion landscape within single ER–mitochondria contact sites. a**, Ensemble image of ER and mitochondria. Arrow indicates ERMCS (one cell of 24 similar shown). **b**, VAPB trajectories within this ERMCS (pseudo coloured by molecule). **c**, Time traces of selected trajectories in **b**. **d**, Measured dwell time of VAPB molecules at all ERMCSs measured. Inset shows leaving frequency of molecules within the ERMCS ($n$ = 427 binding interactions, median = 556 ms). **e**, VAPB localization density of an ERMCS with representative interacting single VAPB trajectory colour coded by diffusion state. **f**, Distance of single VAPB molecule from the ERMCS centre (pseudo coloured by diffusion state, arrows indicate free ER diffusion). **g**, Effective one-dimensional diffusion coefficient of trajectory segments in ER tubules versus in ERMCSs ($n$ = 700 segments in free ER, 48 segments in ERMCSs; $P$ < 0.0001, two-sided Mann–Whitney, error bars show 95% confidence interval of the median). **h**, Representative centre-aligned trajectory segments of free VAPB, ERMCS-associated VAPB or immobilized beads. **i**, Localization density of two adjacent ERMCSs on the same mitochondrion (CS1 and CS2). Lines represent axes used in **k**. **j**, Location of single-molecule steps associated with the contact sites in **e**, coloured by mean two-dimensional (2D) $D_{eff}$ in the local neighbourhood. **k**, Mean 2D $D_{eff}$ in a 30 nm neighbourhood at each 10 nm step along the lines specified in **e**. Colourmap indicates net localization density. **l**, 3D EM reconstruction of an ERMCS. **m**, ER membrane within the contact site colour coded by local curvature. Arrows indicate sites with negative ER curvature within potential tethering distance for VAPB. **n**, Collective VAPB localization density across aligned contact sites between ER tubules and mitochondria in the cell periphery. Red bar indicates approximately Gaussian decay at the contact site centre, grey bar indicates shoulders extending into the ER tubules. Scale bars, 5 μm (**a**), 500 nm (**b**,**e**,**f**,**i**,**j**), 2 μm (**h**). Time bar, 1 s (**f**).

cells highly overexpressing both VAPB and PTPIP51 showed essentially complete depletion of the ER pool of VAPB, with the protein instead enriched at expanded contact sites (Fig. 3g). Thus, the availability of mitochondrial tethers controls VAPB-mediated tethering between ER and mitochondria.

We next explored how ERMCS structure and dynamics adapt to physiological challenges for meeting cellular needs. Prior biochemical and cryo-EM studies have suggested ERMCSs expand and change composition during acute nutrient deprivation[11,34–37]. These changes are proposed to provide crucial lipids and calcium to mitochondria to fuel oxidative phosphorylation or stimulate apoptosis[38,39]. To investigate

this expansion and its effects on contact site substructure, we employed our sptPALM approach. VAPB trajectories were used to measure the size and organization of contact sites in COS7 cells after 8 h of acute nutrient deprivation, achieved by culturing the cells in Hanks' balanced salt solution (HBSS). A significant expansion in the size of contact sites was observed (Fig. 3h–j). Mapping the diffusion landscape of VAPB in contact sites within starved cells, we observed that they still contained a single central low diffusion subdomain despite their expanded size and shape (Fig. 3k,l and Extended Data Fig. 8a–c). The effective diffusion coefficient of VAPB within the central subdomain, however, was significantly lower than that measured in contact sites from well-fed

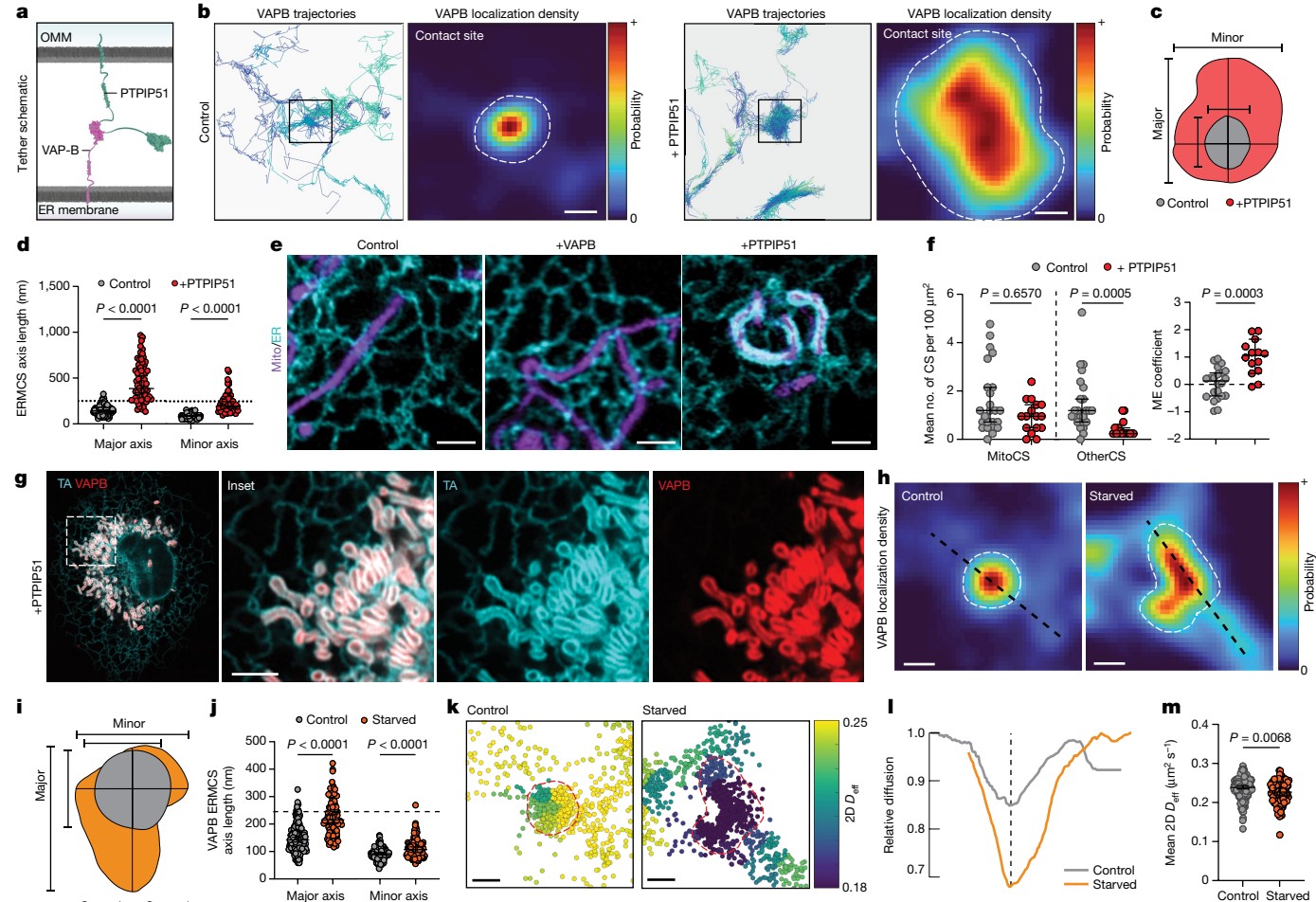

**Fig. 3 | VAPB contact sites dynamically reorganize according to tether availability and metabolic needs. a**, Cartoon of VAPB-PTPIP51 tethering at an ERMCS. **b**, Representative trajectories and corresponding localization densities for VAPB at ERMCSs in control (left) or PTPIP51-overexpressing (right) cells. Dashed lines indicate contact site boundaries. **c**, Overlaid ERMCS footprints from **b** with axes indicated. **d**, VAPB-associated ERMCS size in control or PTPIP51-overexpressing cells ($n = 160$ ERMCSs (control), 64 ERMCSs (+PTPIP51); major $P < 0.0001$, minor $P < 0.0001$; Dunn's multiple comparisons test, two-sided). Line indicates confocal microscopy resolution limit. **e**, Airyscan micrographs of ER (cyan) and mitochondria (magenta) in either control, VAPB-overexpressing, or PTPIP51-overexpressing cells. **f**, Number of total VAPB organelle contact sites in control or PTPIP51-overexpressing cells ($n = 24$ cells (control), 16 cells (+PTPIP51); $P = 0.6570$, $P = 0.0005$; Dunn's multiple comparisons test, two-sided) and mitochondrial enrichment coefficient derived from the same cells ($n = 22$ cells (control), 14 cells (+PTPIP51); $P = 0.0003$; Dunnett's T3 multiple comparisons test, two-sided). **g**, Airyscan micrograph of TA (cyan) and VAPB (red) in a COS7 overexpressing PTPIP51 (untagged). **h**, Representative VAPB localization density at ERMCSs in control (complete medium) or starved (HBSS 8 h) cells. Line profiles indicate axes used for **i**; dashed lines indicate contact site boundaries. **i**, Overlaid ERMCS footprints from **h**. **j**, Contact site size in control or starved cells ($n = 160$ ERMCSs (control), 96 ERMCSs (starved); major $P < 0.0001$, minor $P < 0.0001$; Dunn's multiple comparisons test, two-sided); line indicates confocal microscopy resolution limit. **k**, Single-molecule steps associated with the contact sites in **h**, coloured by the mean 2D $D_{eff}$ in the local neighbourhood. **l**, Effective 2D diffusion coefficient of VAPB from **h**, normalized to neighbouring ER regions (dashed line indicates contact site centre). **m**, Mean 2D $D_{eff}$ within contact sites in either control or starved cells ($n = 160$ ERMCSs (control), 96 ERMCSs (starved); $P = 0.0068$; Dunn's multiple comparisons test, two-sided). Error bars indicate 95% confidence interval of the median. Scale bars, 200 nm (**b**,**h**,**k**), 2.5 μm (**e**), 2.5 μm (**g**).

cells, despite continued VAPB interaction with both ERMCSs and contact sites with other organelles (Fig. 3l,m and Extended Data Fig. 8d,e). These results indicated that even though ERMCSs become larger in starved cells, they maintain their dynamic properties. VAPB molecules are still highly enriched in central subdomains and retain the ability to move in and out of the contact site rapidly. This dynamic remodelling of the ERMCS into an expanded interface under starvation likely enables the ER and mitochondria to engage in more efficient metabolite exchange for cell survival.

Subsequently, we examined whether disease-causing forms of VAPB affected contact site organization. Several mutations in the gene encoding VAPB are causative for the motor neuron disease amyotrophic lateral sclerosis (ALS)[18]. In patients, ALS is most clearly associated with hyper-functional mitochondria and associated oxidative stress[40,41]. Disease-causing VAPB mutations generally encode more aggregation-prone versions of the protein and confer disease as a dominant, highly penetrant allele[42]. Aggregated VAPB molecules are non-functional[43–46] and degrade quickly[47], so disease states have been proposed to be conveyed through toxic effects of aggregates or by gene dosage effects due to aggregated proteins being degraded[42]. An additional possibility, however, is that before becoming immobilized in aggregates and eventually degraded, mutant VAPB molecules in the ER target to ERMCSs and cause aberrant contact site dynamics and/or function.

To distinguish between these possibilities, we tracked the motion of single VAPB molecules carrying the well-characterized ALS8 missense mutation (P56S) found in the MSP domain responsible for mitochondria interaction (Fig. 4a)[14,48]. In agreement with results from biochemical

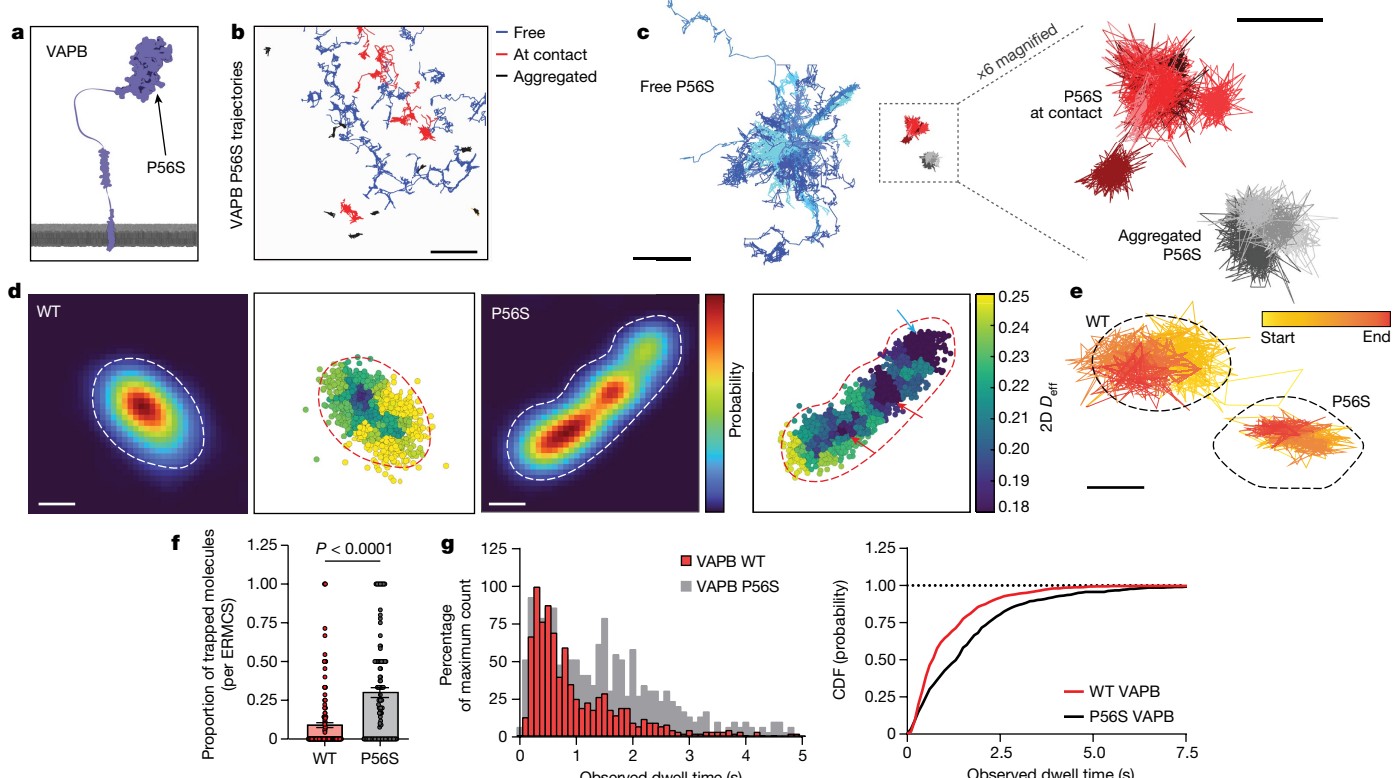

**Fig. 4 | The ALS-linked VAPB P56S mutation displays aberrant motility at ER–mitochondria contact sites. a**, A cartoon indicating the approximate location of the P56S mutation in the MSP domain of VAPB. **b**, Selected molecular trajectories of P56S VAPB in a representative COS7 cell. Tracks are colour coded by their primary state of motion (Methods). **c**, Grouped and overlaid representative trajectory segments of P56S VAPB in COS7 cells. **d**, Localization density in a representative ERMCS of HaloTag fused to either WT or P56S VAPB and corresponding local diffusion maps. Note the existence of multiple distinct low diffusion wells in the P56S VAPB ERMCS landscape, some of which correspond to high net tether density (red arrows) and some of which do not (blue arrow). Dotted lines delineate the edge of the contact site.

**e**, Single representative ERMCSs of WT or P56S VAPB with the trajectory traversed by a single VAPB molecule plotted above. Note the effective confinement of P56S VAPB resulting in a trapped state, the molecule does not encounter the edges of the ERMCS. **f**, Quantification of the mean proportion of molecules trapped in WT or P56S VAPB ERMCSs for the full lifetime of the fluorochrome. ($n = 160$ (WT), 120 (P56S) contact sites, bars show mean ± s.e.m.; $P < 0.0001$, two-sided Mann–Whitney test). **g**, Observed dwell times for WT or P56S VAPB molecules in ERMCSs and the associated cumulative distribution function for trajectories in the bound state ($n = 734$ (WT), 483 (P56S) binding interactions; median values: WT, 715 ms; P56S, 1.358 s). Scale bars, 5 μm (**b**), 2 μm (**c**, left), 500 nm (**c**, right), 200 nm (**d**), 250 nm (**e**).

studies describing insoluble ER-associated aggregates[19,43,49], many P56S VAPB molecules were in an immobilized form (Fig. 4b,c, grey), rarely changing to other dynamic states. A significant fraction of P56S VAPB molecules also showed signatures of free ER diffusion like wild-type (WT) VAPB (Fig. 4b,c), indicating that some P56S VAPB molecules are still effectively inserted into the ER in an unaggregated form. Notably, however, an additional pool of P56S VAPB molecules showed distinct signatures of contact site-associated motion (Fig. 4b,c). This demonstrated that some P56S VAPB molecules still reached ERMCSs and raised the possibility that P56S VAPB might impair ERMCS structure and/or dynamics.

To assess whether P56S VAPB molecules impact ERMCSs, we compared the properties of contact sites derived from trajectories of WT VAPB or P56S VAPB. Although P56S VAPB-containing contact sites often qualitatively looked very similar to WT VAPB contact sites, they showed a slight but statistically significant increase in size and decrease in effective diffusion (Extended Data Fig. 9a–d). However, examination of ERMCS subdomain architecture and VAPB dynamics at the nanoscale revealed striking differences. P56S VAPB-containing contact sites, unlike their WT counterparts, often showed multiple distinct low diffusion subdomains (Fig. 4d and Extended Data Fig. 9e,f), some of which were surprisingly not correlated to tether density at all. Thus, these subdomains unlikely represent the metastable structures universally observed in WT contact sites. Here, P56S VAPB molecules appeared

trapped, showing a high degree of confinement within the subdomain rather than exploring the entire ERMCS (Fig. 4e). Consequently, a significant fraction of tracked P56S VAPB molecules remained in the ERMCS for the entirety of the trajectory lifetime (Fig. 4f), a result of their impaired capacity to reach the contact site edges and escape from the ERMCS. Indeed, the observed dwell time of P56S VAPB molecules in ERMCSs was significantly longer than that of WT VAPB molecules, suggesting P56S VAPB expression causes a dramatic change in the overall organization and plasticity of the ERMCS interface.

## Discussion

ERMCS are central hubs for cellular metabolism that regulate lipid and metabolite exchange between ER and mitochondria. Here we provide direct observations of the behaviour and organization of tethering machinery comprising these crucial structures in a living cell. Existing models of interorganelle contact sites generally depict protein complexes that tether stably together for many seconds. Indeed, prior work examining the dynamism of molecules at ER–plasma membrane contact sites have reported dwell times of tens of seconds to minutes[50], suggesting an overall pseudo-stability. Our results using both high-speed molecular tracking of VAPB tethers and structural volumetric FIB-SEM paint a different picture of ERMCSs. ERMCSs exhibit exceptional plasticity, with dwell times of VAPB approximately ten times more transient

than those reported for proteins at ER–plasma membrane contact sites. Despite this dynamism, ERMCSs maintained spatially organized subdomains in the steady state, enabling contact site remodelling in response to starvation. Underscoring the importance of this metastability, we showed disease-causing mutations in VAPB disrupt this architecture and plasticity. The unprecedented dynamic exchange of components and exquisitely maintained nanoscale structure that we report here present a new paradigm for understanding ERMCSs and will help direct future work aimed at characterizing the biology and function of contact sites.

A key implication of these findings is that interactions between tethers likely consist of many rapid binding and unbinding events across the entire structure of the contact site, consistent with the known low affinity of VAPB for FFAT-containing binding partners like PTPIP51 (refs. 14,17,51). This results in a gradual increase in the VAPB abundance across the contact site with a clear peak at the centre, where the likelihood of rebinding events would be elevated due to an increased density of potential tethering molecules. Additionally, this higher abundance of tethers towards the centre of the contact site would be predicted to provide increased adhesive force between the two membranes, a phenomenon we could directly observe as regions of negative curvature in the centre of ERMCSs in FIB-SEM data sets. The surrounding edge regions of the contact site could serve as important staging sites for other ERMCS-associated activities. Supporting this idea, the location of ERMCSs directly correlated to the location of mitochondrial cristae and sites of constriction in the mitochondrial membrane, biological phenomena that require recruitment of specific cellular machinery in addition to tethers[25–27,52,53].

Our results further revealed ERMCS interfaces exhibited plasticity, being able to adjust their structure and organization to different conditions. The size and shape of contact sites, for example, could be regulated through the availability of tethering partners like PTPIP51. Moreover, they became larger and exhibited reduced VAPB mobility during nutrient deprivation, which likely allows more efficient interorganelle metabolite transfer, as has been previously reported in biochemical studies of cells undergoing acute starvation[34–37]. One way these contact site transformations might occur physiologically is through regulation of either VAPB–PTPIP51 interactions or lateral associations between VAPB molecules within the ER via posttranslational modification of either component (as suggested at ER–plasma membrane contacts, see Supplementary Text, section 2c, for discussion). Indeed, there is biochemical data suggesting posttranslational modifications of both components are precisely regulated[14,17,51,54,55]. The dynamic exchange of VAPB molecules we observe between ERMCS and the surrounding ER suggests that posttranslational modifications to VAPB need not occur within the contact site, since the ERMCS- and ER-localized pools will rapidly equilibrate. Access to the spatially confined ERMCS environment would likely provide steric challenges for the bulky phosphatases and kinases implicated in VAPB regulation. We suspect that this strategy could be a generic regulatory mechanism, applying to other tethers and contact sites.

The importance of the dynamic exchange of VAPB at ERMCSs was underscored by our high-speed tracking of P56S VAPB. Although most P56S VAPB molecules were aggregated or undergoing free ER diffusion, a small but significant portion were still at ERMCSs. Notably, these ERMCS-associated P56S VAPB molecules disrupted normal VAPB diffusion in the contact site, with many molecules becoming trapped in multiple small subdomains of the ERMCS. We speculate that a failure of these trapped molecules to leave the ERMCS may lead to an impaired ability of the contact site to undergo normal dynamic restructuring, either through altered PTPIP51 interactions or by changes in dimerization/lateral aggregation of VAPB within the contact site (see Supplementary Text, section 2e, for discussion). More stable interfaces between ER and mitochondria have been shown in a variety of contexts to result in elevated mitochondrial function due to ER-to-mitochondria

signalling[11,34,54,56]. Thus, even a small fraction of P56S in ERMCSs could directly lead to corrupted ERMCS signalling and the oxidative stress known to occur in VAPB-associated ALS disease[40,54].

In summary, our results demonstrate that ERMCSs have distinct subdomains and a profoundly dynamic nature, with tethers diffusing into and out of the site over time scales of milliseconds and contact site size and configuration changing in response to different stimuli. Future work combining FIB-SEM structural data with single particle tracking promises to yield further insight into ERMCS biology, as well as impact understanding at the molecular level of other dynamic interfaces involved in interorganelle communication.

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

# Methods

## Coverslip cleaning and preparation

For single-molecule imaging, high tolerance, Number 1.5 coverslips were purchased from Warner scientific (25 mm) and precleaned with a modified version of a previously described protocol[57,58]. Coverslips were sonicated overnight (approximately 12 h) in 0.1% Hellmenex II (Sigma), followed by five washes in 300 ml of distilled water. Coverslips were then transferred to a clean chamber of 300 ml of distilled water and sonicated overnight again, followed by five more washes. Coverslips were then ethanol sterilized in pure, 200 proof ethanol and air dried in a clean tissue culture hood. After cleaning, coverslips were stored in an airtight container until use, and were discarded if not used within 30 days of cleaning.

For FIB-SEM, sapphire coverslips (3 mm diameter, 50 μm thickness, Nanjing Co-Energy Optical Crystal Co. Ltd, COE) were cleaned for at least 1 h in a basic piranha solution (5:1:1 water:ammonium hydroxide:hydrogen peroxide) followed by several washes in distilled water. The bottom of each coverslip had a thin layer of gold sputtered onto the side regions of the coverslip to distinguish top from bottom in subsequent steps (sputter coater Desk II, Denton Vacuum). After sputtering, coverslips were rinsed several times in distilled water and stored under vacuum in a desiccation chamber until use.

## Plasmids and reagents

ER-mRFP (Addgene no. 62236), mTagRFP-T2-Mito-7 (Addgene no. 58041) (referred to as mitoRFP in the text), mTagBFP2-N1 (Addgene no. 54566), mEGFP-N1 (Addgene no. 54767), mEGFP-C1 (Addgene no. 54759) and mEmerald-Sec61b-C1 (Addgene no. 90992) have been described previously, and were gifts from Erik Snapp, Michael Davidson, or generated in house. EGFP-VAPB[59], HA-PTPIP51 (ref. 16) and pHAGE-Tet-STEMCCA[60] have been previously described and were gifts from Pietro De Camilli, Kurt De Vos and Robert Tijan, respectively.

All insertions and cassette changes were performed using the NEBuilder implementation of Gibson Assembly (New England Biolabs) unless specified otherwise, taking care to leave appropriate restriction sites for later changes. All constructs were sequenced before use and will be available on Addgene unless prohibited by copyright. Specific strategies and resulting plasmid maps are linked in the Supplementary Information.

## Cell culture and transfection

COS7 cells were purchased from ATCC and used within 40 passages. Cells were maintained in complete DMEM (phenol red-free Dulbecco's modified Eagle medium (Corning) supplemented with 10% (volume/volume) fetal bovine serum (Corning), 2 mM L-glutamine (Corning), 100 U ml$^{-1}$ penicillin and 100 μg ml$^{-1}$ streptomycin (ThermoFisher)). Cells were cultured at 37 °C in 5% $CO_2$, passaging was accomplished with 0.25% (weight/volume) trypsin (Corning) and care was taken never to let cells grow to more than 85% confluency or be seeded at less than 25% confluency, as they often become less flat after this.

For single-molecule imaging, coverslips were precoated with 500 μg ml$^{-1}$ phenol red-free Matrigel depleted for growth factors (Corning) for 1 h before plating in a 35 mm tissue culture dish. Cells were seeded to ensure less than 60% confluency at the time of imaging to maximize regions of ER within the focal plane of the objective when focused just above the coverslip. Transfections were performed after letting the cells adhere to the glass for at least 12 h using Fugene6 (Promega) at a 3:1 Fugene (μl) to DNA (μg) ratio according to the manufacturer's protocol. Each 35 mm dish was transfected with 1.5 μg of DNA using the following ratio: 750 ng PrSS-mEmerald-KDEL (ER structure label), 500 ng mitoRFP (mitochondria structure label) and 250 ng of the HaloTag construct used for sptPALM, except for cells transfected with PTPIP51 which were given an extra 500 ng of PTPIP51-IRES-mTagBFP2. Imaging was always performed at least 12 h after transfection, but always before 24 h and taking care to avoid cells showing morphological changes from ER and mitochondria label expression that become increasingly common at later time points. For starvation experiments, cells were incubated in HBSS for the last 8 h before imaging, but complete medium was replaced on the cells immediately before imaging.

Cells prepared for Airyscan imaging were plated in high tolerance commercially acquired 35 mm imaging chambers (MatTek Life Sciences). Briefly, coverslips were precoated with 500 μg ml$^{-1}$ phenol red-free Matrigel depleted for growth factors (Corning) for 20 min. Cells were then transfected in solution using Fugene6 (Promega) according to a modified version of the manufacturer's protocol. Briefly, cells were resuspended after pelleting in prepared transfection complexes made according to the manufacturer's recommendations in OptiMEM (ThermoFisher). Cells were incubated for 15 min at 37 °C before being plated on the coated coverslips in 2 ml of complete Medium. Imaging was performed 18–24 h after transfection.

## Halo labelling

Cells were labelled for sptPALM by incubating with 10 nM PA-JF646 (ref. 29) in OptiMEM (ThermoFisher) for 1 min followed at least five washes with 10 ml of PBS, performed while simultaneously aspirating and taking care to never let the cells come in direct contact with the air. The cells were then washed once with 10 ml of complete DMEM and left to recover in 2 ml of complete DMEM for 15 min before imaging.

Cells were labelled for Airyscan imaging by replacing the complete medium on the cells with complete DMEM supplemented with 10 nM JF635 (ref. 61). The highly fluorogenic nature of this JaneliaFluor (JF) dye compound removes the need for washing steps, and the sample can directly be imaged on the microscope.

## Microscopy and imaging conditions

Single-molecule imaging was performed using a custom widefield microscope assembled in an inverted Nikon Ti-E outfitted with a stage top incubator to stabilize temperature, $CO_2$, and relative humidity during imaging (Tokai Hit). The flat lamella of cells where sptPALM is possible (approximately 500 nm thick or less) were located using eye pieces to visualize the ER and mitochondria localization. To avoid bias, the experimenter was always blinded to the single-molecule tracers. In experiments where PTPIP51-IRES-mTagBFP2 was overexpressed, the cells were selected using the fluorescence of the mTagBFP2 in addition to the ER and mitochondria structure.

Excitation was performed using three fibre-coupled solid state laser lines (488 nm, 561 nm, 642 nm; Agilent Technologies) introduced into the system with a conventional rear-mount TIRF illuminator. The angle of incidence was manually adjusted for each cell beneath the critical angle to maximize the evenness of the illumination in the ER. The illumination in the 488 nm and 561 nm channel was manually adjusted based on the brightness of the sample to minimize fluorescent bleed-through, but the total power on each line was always kept less than 50 μW and 150 μW total in the back aperture, respectively. Single molecules were always imaged using a constant total power of 11.5 mW of 647 nm light in the back aperture. If necessary, a small amount of 405 nm light was introduced to tune the photoactivation rate of the molecules being tracked, but in practice this was rarely needed.

Emitted light was collected with a ×100 α-plan apochromat 1.49 numerical aperture oil immersion objective (Nikon Instruments) and focused through a MultiCam optical splitter (Cairn Research). The emission path was split onto three arms of the splitter using a 565LP and a 647LP dichroic mirror (Chroma) placed sequentially in the optical path to split the light from the 488 nm and 561 nm channels, respectively. These emission paths were additionally cleaned up by passing the emitted light through a 525/50 BP and a 605/70 BP filter (Chroma), respectively. The remaining light transmitted through the MultiCam represents the far-red signal where the single molecules of HaloTag-linked dye are imaged, and the path was passed through

an additional 647LP filter to clean up any stray light in the system that could decrease the resolving power of the sptPALM approach. All three channels were collected from electronically synchronized iXon3 electron multiplying charged coupled device cameras (EM-CCD, DU-897; Andor Technology). To image quickly enough, the field of view was reduced to a 128 × 128 pixel square (20.48 µm × 20.48 µm). The location of the square was carefully chosen for each sample to contain the flattest region of ER possible while remaining near the centre of the camera chip, since the objective in use is chromatically corrected to high precision only near the centre of the field of view. Imaging was performed with 5 ms exposure times for 60–90 s at a time, and the timing of each frame was monitored using an oscilloscope directly coupled into the system (mean frame rate of approximately 95 Hz).

Airyscan imaging was performed using a commercially acquired Zeiss LSM 880 microscope with a live cell incubation system (Zeiss Microscopy). Briefly, labelled samples were sequentially excited with laser lines at 633 nm, 561 nm and 488 nm. Emission fluorescence is collected using a ×63 1.4 numerical aperture oil immersion objective (Zeiss Microscopy) with an open pinhole and passed through an appropriate custom bandpass filter based on the expected emission profile of the sample to the arrayed detector for the Airyscan unit (561 nm, 488 nm, BP495-550 + LP570; 633 nm, BP570-620 + LP645). Airyscan reconstruction and deconvolution was performed using the default settings (filter size = 6). Images were pseudocoloured and prepared using Fiji (NIH) for visibility.

## Channel registration and spectral analysis

At the time of imaging, a crude channel alignment was performed using a sparse distribution of Tetraspek beads on a coverslip, prepared and imaged as for other sptPALM samples. The angle of the dichroic mirrors was manually adjusted to get as much overlap between the channels in the main field of view as possible. In practice, this alignment was sufficient to support the manual steps in the tracking pipeline (see below), but applications requiring more precise alignment were accomplished using a custom subpixel alignment pipeline in Fiji.

Variation in expression level of the three markers (ER structure marker, mitochondria structure marker and single-molecule tracer) due to uneven transfection is not a trivial issue, and often required manual adjustment in the relative laser power for the 488 nm and 561 nm lines by the experimenter. Since this could in principle create artefacts in the automated analysis pipeline or introduce erroneous single-molecule localizations as a result of bleed-through, all of the samples were run through an automated spectral analysis pipeline that checked for fluorescence contamination from the blue-shifted channels. Any samples where the detectable bleed-through contamination was significant compared to the signal of single molecules were removed before downstream analysis was performed.

## Localization and tracking

Localizations were identified in the single-molecule data sets using a previously described pipeline[62] to estimate positions and precision of localization using a maximum likelihood estimation (MLE)-based fitting approach. The quality filter used in the downstream tracking pipeline limited analysable localizations to those identified with precision (as estimated from the Cramér–Rao lower bound) in the range of 20–30 nm.

Trajectories were assembled from single-molecule images using the TrackMate plugin in Fiji[63,64]. Linking parameters were experimentally selected for each dataset to minimize visible linkage artefacts as determined by eye. Resulting putative trajectories were then projected on to the simultaneously collected ER network structure, and manually curated to remove any trajectory linkages that are close in 2D but far from one another in the underlying organelle structure. This step proved crucial to assembling trajectories that moved within the structure without linkage artefacts. Resulting trajectories were exported from TrackMate and imported into MATLAB for subsequent analysis.

## Spatial density analysis and contact site identification

Spatial probability density is mapped by choosing the spatiotemporal boundaries of the data to be analysed (x, y, t) and binning the resulting localizations into 30 nm square pixels. The resulting counts are normalized to the total number of localizations within the dataset, and as such probability represents solely the likelihood of a single molecule falling in a certain pixel if chosen at random (that is, a spatially defined probability mass function). This effectively minimizes the effects of differences in photoactivation efficiency or tagged protein expression level when identifying the boundaries of a contact site. Note that this analysis does not assume anything about the motion of the trajectories, the orientation or stability of the contact site, or the nature of molecular interactions—all of this information is analysed in subsequent steps.

The initial location of contact sites was identified from the spatially defined probability density when calculated over the entire image, but the location and boundaries often had to be manually refined, especially under conditions where contact sites move or change orientation (Extended Data Fig. 5).

## Spatial clustering and diffusion landscape estimation

To generate a map of the diffusion landscape within contact sites, the space inside the contact site was divided into distinct compartments by Voronoi tessellation informed by the probability density at the site. The trajectories associated with the site were broken into single steps and assigned to a tessellation by the location of the beginning of the step[32,33]. Bayesian inference was then used to model the resulting distribution as an overdamped Langevin system within each tessellation, assuming single molecules in the same space at distinct times can be viewed as independent experiments probing the same molecular environment (see Supplementary Text, section 6c). The resulting diffusive component was reported for each tessellation as an effective 2D diffusion coefficient.

## Identification of latent states in single trajectories

All trajectories longer than 500 steps (approximately 5.5 s) were analysed using a non-parametric Bayesian modelling technique (Hierarchical Dirichlet Process Modelling, HDP) coded using Python to estimate latent state changes in single-molecule behaviour[30,31]. Briefly, the system was treated as a switching linear dynamical system (SLDS). As in previous work[30,31], an overdamped Langevin equation was used to interpret the parameters of the linear dynamical system used in the SLDS model. This approach removes the need for an upper bound on the number of potential states common in single particle tracking analysis approaches (Hidden Markov Models, and so on), which become intractable in a spatially complex environment like the ER (Supplementary Text, section 6d). Eigen-decompositions of the implied force and diffusion tensors for each determined state enables one-dimensional analysis through tensor diagonalization. Note this SLDS treats thermal fluctuations as a distinct component from measurement noise, allowing diffusive properties of the system to be quantitatively estimated independently of measurement noise (for example, localization errors).

## High pressure freezing and freeze-substitution

Immediately before freezing, cells were manually inspected using an inverted widefield microscope to ensure reasonable morphology and viability. Cells were then transferred to a water jacketed incubator (ThermoFisher, Midi 40) where they were kept at 37 °C in 5% $CO_2$ and 100% humidity until ready for freezing. Each sapphire coverslip was removed one at time from the incubator, overlaid with a 25% (weight/volume) solution of 40,000 MW dextran (Sigma), loaded between two hexadecane-coated freezing platelets (Technotrade International),

and placed in the HPF holder. Freezing was then performed using a Wohlwend Compact 2 high pressure freezer, according to the manufacturer's protocol. Frozen samples were stored under liquid nitrogen until freeze-substitution was performed.

Freeze-substitution was performed with a modified version of a previously described protocol[22,65]. Briefly, frozen samples were transferred to cryotubes containing freeze-substitution media (2% OsO$_4$, 0.1% Uranyl acetate and 3% water in acetone) and placed in an automated freeze-substitution machine (AFS2, Leica Microsystems). A freeze-substitution protocol was used as previously described[23], and the samples were then washed three times in anhydrous acetone and embedded in Eponate 12 (Ted Pella, Inc.). The sapphire coverslip was then removed and the block was re-embedded in Durcapan (Sigma) resin for FIB-SEM imaging.

### FIB-SEM

FIB-SEM was performed essentially as previously described[21,23,66]. Briefly, a customized FIB-SEM using a Zeiss Capella FIB column fitted at 90 degrees on a Zeiss Merlin SEM was used to sequentially image and mill 8 nm layers from the Durcapan-embedded block. Milling steps were performed using a 15 nA gallium ion beam at 30 kV to generate two sequential 4 nm steps. Data was acquired at 500 kHz pixel$^{-1}$ using a 2 nA electron beam at 1.0 kV landing energy with 8 nm $xy$ resolution to generate isotropic voxels. Data sets were registered postacquisition using a SIFT-based algorithm[67].

### Voxel classification and surface determination

Although several automated segmentation protocols exist for reconstruction of organelles from FIB-SEM data[10,21], we found that minor errors in voxel classification within the contact sites themselves obscured our ability to analyse the local curvature in sufficient resolution for our needs (see Supplementary Text, section 3a, for discussion). Consequently, we selected a few small volumes containing mitochondria and manually classified the voxels for the ER using Amira (ThermoFisher). We used a modified watershed algorithm to classify the mitochondrial membranes but performed a manual curation to remove artefacts. Potential contact sites on the ER surface were identified as regions of ER membrane within 24 nm (approximately three pixels) of the OMM, as measured by dilation of the OMM (see Supplementary Text, section 3b, for discussion of contact site distances).

### Triangulation, smoothing and curvature analysis

Triangulated surfaces were fit to the voxel classifications using a marching cubes-based algorithm implemented directly in Amira. To avoid voxel-step artefacts in the surface, gaussian smoothing was applied to the voxel data using a local likelihood measure selected over a kernel size relevant for the expected curvature of the underlying membrane (Supplementary Text, section 3c). The resulting triangulated surfaces were rendered for use in the figures using Amira, and they serve as the scaffold for subsequent 3D curvature analysis.

Mean local curvature of the ER was computed as a scalar field over the triangulated surface using a 20-layer neighbourhood to fit a quadratic form along the two principal curvature axes. Note this is different from gaussian curvature, and the resulting value is negative in strictly concave regions and positive in regions that are strictly convex. Scalar fields were calculated and mapped using the curvature field module in Amira. Details are given in the Supplementary Text, section 3d–e.

### Statistics and reproducibility

Each single-particle tracking dataset contains a total of at least 16 regions (20.48 μm × 20.48 μm) divided over at least two experiments, each selected from a different cell. Not all of these contained mitochondria-associated contact sites, though all had contact sites of some kind (Extended Data Fig. 2). FIB-SEM data was visually examined in three COS7 cells, but all data shown or quantified in the paper comes from a single representative cell. Single contact site analysis was performed on each of the hundreds of contact sites analysed, each of which contains anywhere from 1–50 VAPB trajectories. All representative images of correlated sptPALM data throughout the paper are a single image of at 16–24 similar acquired, the number of which is listed in the associated quantifications. ERMCS structures shown from FIB-SEM data are representative examples of 25 similar contact sites, except where explicitly stated to be otherwise, such as unusual examples. Airyscan images are representative of at least 30 cells, and Airyscan experiments were performed five times with multiple combinations of fluorescent labels with indistinguishable results (see Supplementary Text, section 8, for a more complete discussion of reproducibility).

### Reporting summary

Further information on research design is available in the Nature Portfolio Reporting Summary linked to this article.

## Data availability

The experimental data used in this paper are freely available in both raw and processed form at figshare with the identifier: https://doi.org/10.25378/janelia.c.6916543.v1. All data are additionally available from the authors in other formats as needed upon request. Source data are provided with this paper.

## Code availability

Custom code was essential to the conclusions of this paper. All codes generated or used in this paper are available at the following GitHub repository: https://github.com/cjobara/Obara-Nixon-Abell-2023-Nature. Codes used for generating figures or specific visualization are available with the data in the figshare repository linked above.

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

**Acknowledgements** This work is funded by the Howard Hughes Medical Institute through Janelia Research Campus and by the National Institute of Neurological Disorders and Stroke through the intramural research programme. We thank C. Ott, L. Benedetti, P. Sengupta, E. Betzig and P. Zheng for advice on experimental protocols and critical reading of the manuscript. We thank E. Snapp, L. Lavis and the the Janelia Open Chemistry and Tool Translation teams for generously providing reagents and advice on their use. We are thankful to E. Wait for computational support and the providing access to the HIP before public release.

**Author contributions** C.J.O., J.N.-A., C.B. and J.L.-S. conceived of this work and designed the experiments. C.J.O. and F.R. designed and generated all constructs. C.J.O. designed the optical set up and sptPALM experimental pipeline. C.J.O. and J.N.-A. performed all optical imaging experiments. J.N.-A. performed all blinded curation of sptPALM data. C.J.O., J.-B.M. and C.P.C. designed the sptPALM analysis pipelines and wrote all the code used in the manuscript. D.P.H., G.S., C.S.X., K.S. and H.A.P. optimized and performed all FIB-SEM sample preparation, data collection and data alignment. C.J.O. and D.P.H. developed the FIB-SEM analysis pipeline.

C.J.O., J.N.-A. and J.L.-S. wrote the paper. C.J.O. and A.S.M. designed and generated the figures and videos in the paper. H.F.H., C.B. and J.L.-S. supervised the work and guided the paper writing and figure assembly.

**Competing interests** The authors declare the following competing interests: C.P.C. is the founder and employed by Ursa Analytics, a company that offers statistical analysis services such as the non-parametric Bayesian approach utilized in this paper. All other authors declare no competing interests.

**Additional information**
**Correspondence and requests for materials** should be addressed to Christopher J. Obara or Jennifer Lippincott-Schwartz.

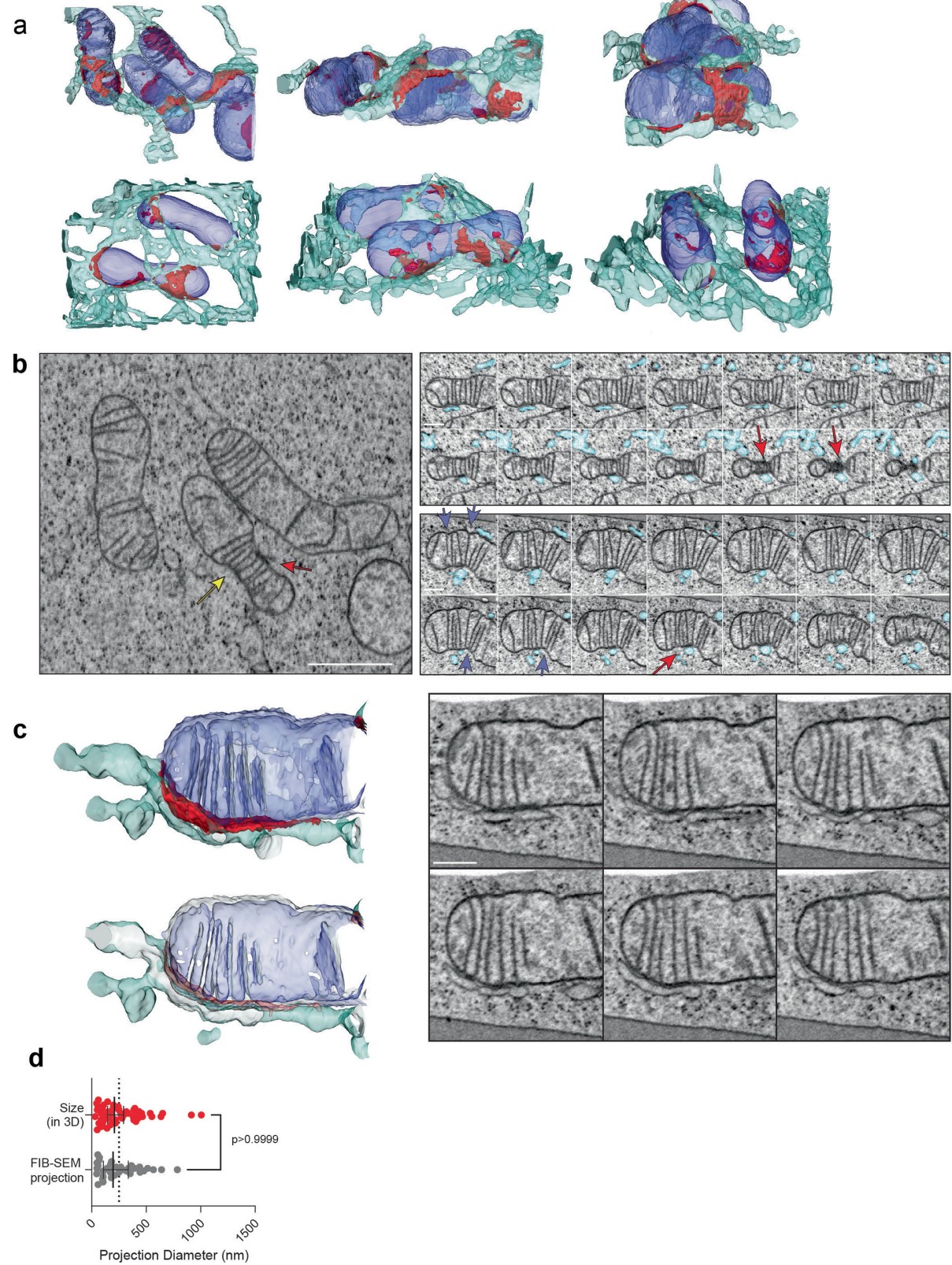

**Extended Data Fig. 1** | See next page for caption.

**Extended Data Fig. 1 | FIB-SEM reveals association of ER-mitochondrial contact sites and known contact site-associated biology. a**, Reconstructed surfaces of ER and mitochondria in association from the FIB-SEM volumes used in this study from several angles. Mitochondria blue, ER cyan, ER membrane within approximate tethering distance of VAPB is colored red. **b**, Serial slices through an ER-associated site of mitochondrial constriction. Upper panels are every 24 nm from above as viewed in the image to the left, lower panels are every 48 nm from the side of the yellow arrow in the direction of the red arrow in the left image. Red arrows correspond to regions of contact site interface with unusual electron density, presumably associated with mitochondrial division machinery. Blue arrows indicate regions of mitochondrial constriction propagating away from the site of direct contact by the ER, generally aligned with enriched cristae at the site. This is one of two ERMCSs with this architecture in the volume (see Extended Data Fig. 7). **c**, The site of aligned cristae shown in Fig. 1b with more context. Lower panel is a cut away of the reconstruction above to the central axis of the mitochondria, showing continuity of the cristae with the inner membrane space. Panels to the right are selected single slices of raw EM data through the contact site shown, taken every 64 nm. This is one representative ERMCS of 25 with similar structures throughout the volume. **d**, Comparison of the measured sizes of ERMCSs in three dimensions from FIB-SEM volumes and the corresponding projection into two dimensions blurred by the resolution of sptPALM. Error bars represent the 95% confidence interval of the median (n = 38 [projected], 52 [three dimensional]; p > 0.9999, Dunn's multiple comparisons test, two-sided). Note some ERMCSs that are close in space are no longer distinguishable when projected down into two dimensions. Scale Bars: b, 1 μm, inset, 400 nm; c, 400 nm.

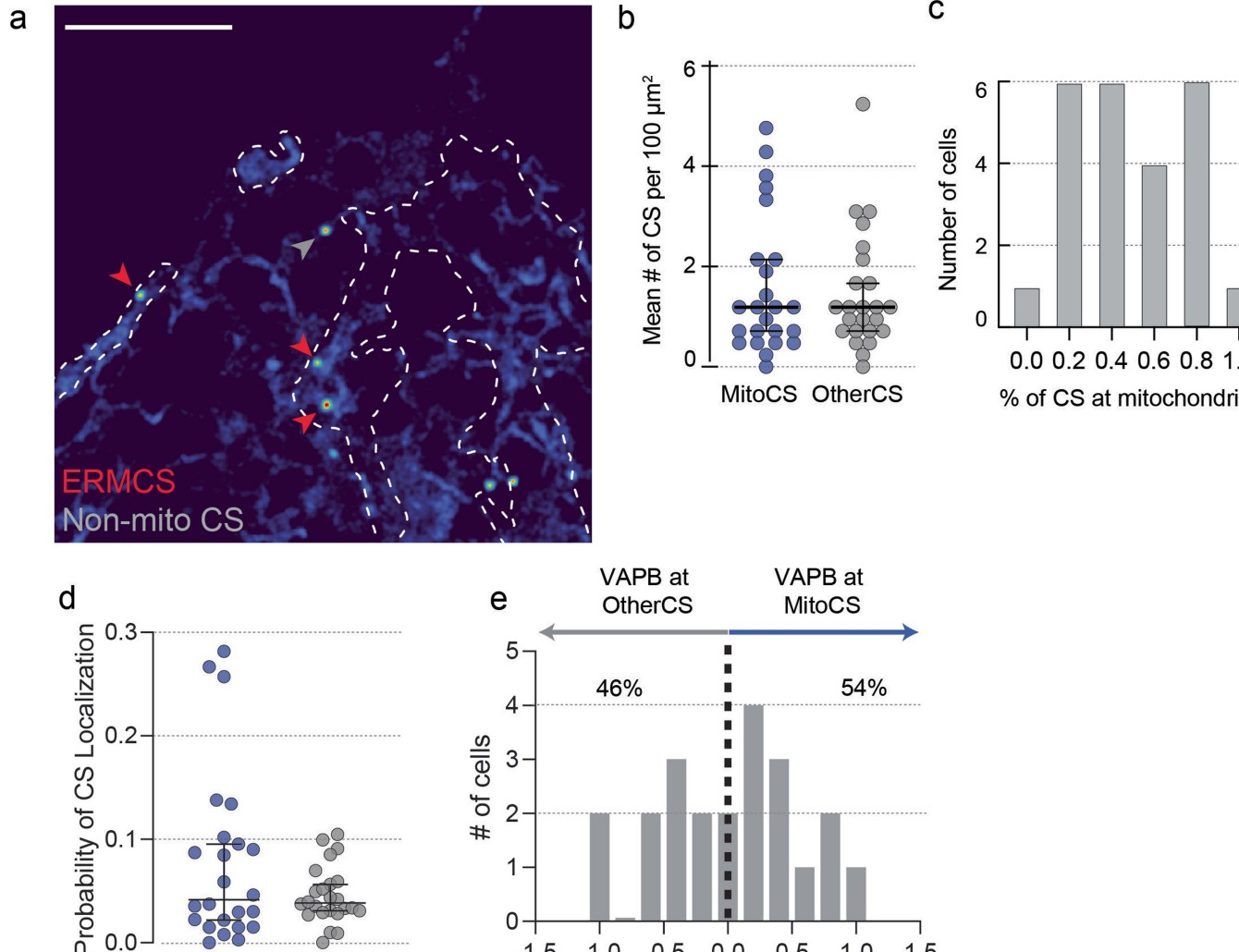

**Extended Data Fig. 2 | Comparisons of VAPB interactions with mitochondria-associated and non-mitochondria associated structures. a**, The likelihood of VAPB localization in a representative cell, showing hotspots associated with mitochondria (red arrows) and a hotspot associated with other structures (grey arrow). Shown is one cell of 24 collected with various hotspot distributions (see Fig. 1 for other examples). **b**, Frequency of VAPB contact sites associated with mitochondria or other organelles in COS7 cells (n = 24 cells from 3 experiments). Error bars indicate the median and 95% confidence interval of the median. **c**, Variability in the proportion of VAPB contact sites that are associated with mitochondria as opposed to other organelles, suggesting adaptability of tether function as a result of cellular needs. **d**, The probability of a randomly selected VAPB molecule localization being associated with contact sites on mitochondria or other structures, as averaged for each cell (n = 24 cells from 3 experiments). Error bars indicate the median and 95% confidence interval of the median. **e**, The distribution of mitochondrial enrichment coefficients over the cells in the dataset, showing heterogeneity in tether targets as a result of cellular needs. Scale bars: 500 nm.

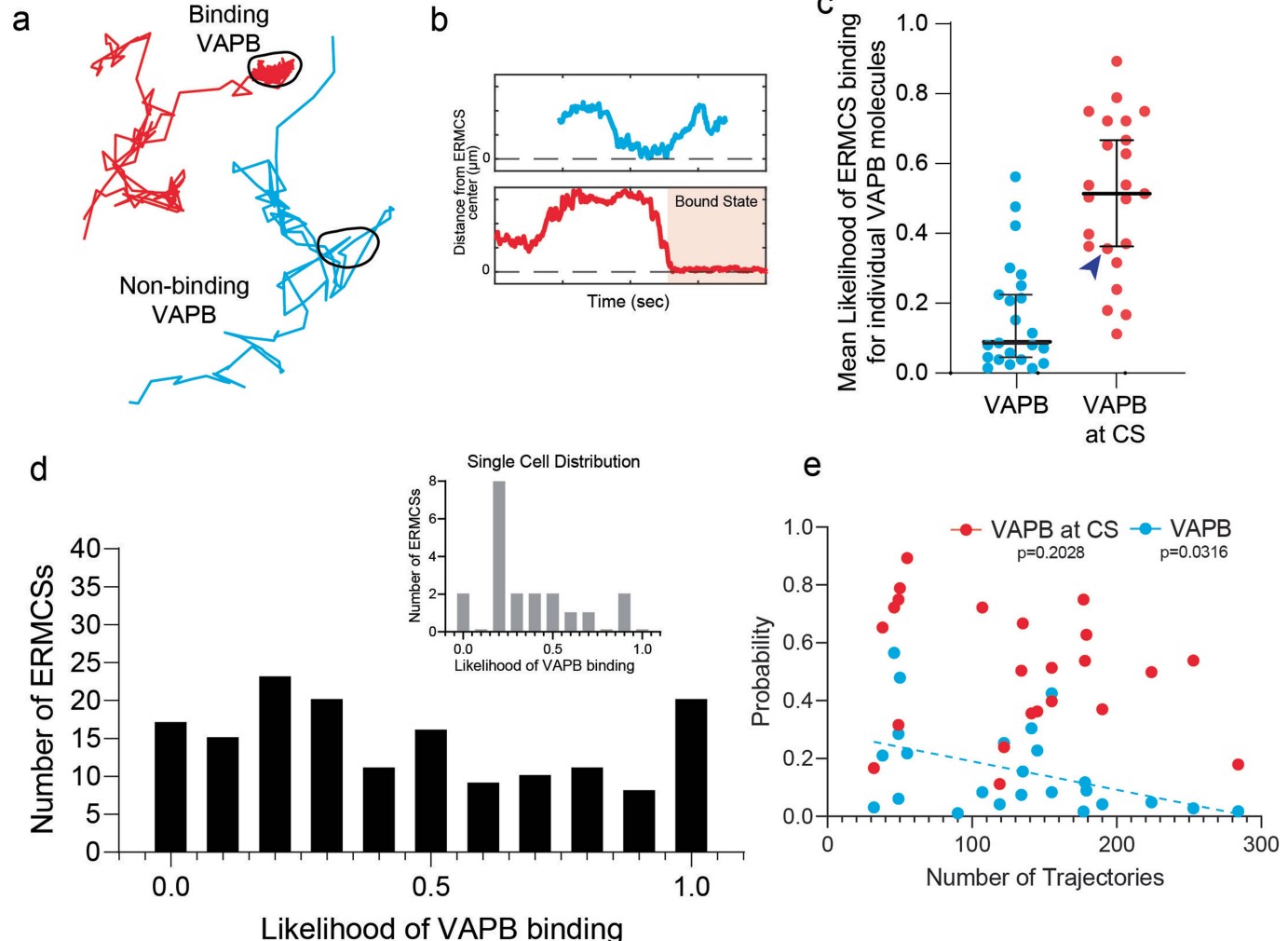

**Extended Data Fig. 3 | Individual ERMCS show variability in capacity to bind passing VAPB molecules. a**, Two representative trajectories of single VAPB molecules passing the same ERMCS. ERMCS boundary is indicated by the black line. **b**, Time traces of the trajectories shown in a; the period of direct contact site interaction is indicated by the shaded background. **c**, The average probability of ERMCS interaction over all VAPB trajectories in a cell or just those that physically encounter an ERMCS, showing the majority of VAPB is freely diffusing in the ER (n = 23 cells). Error bars indicate the median and 95% confidence interval of the median. The blue arrow indicates the cell used as an example in the inset of (d). **d**, The distribution of likelihoods for VAPB binding, as calculated for individual contact sites (n = 160 contact sites). The inset shows the distribution of the single cell indicated by the blue arrow in c (n = 20 contact sites). **e**, Average likelihood of individual VAPB molecules engaging with ERMCSs as a function of VAPB expression level. Note that the probability of ERMCS-associated VAPB engagement is uncorrelated to VAPB expression level, suggesting the law of mass action is satisfied (n = 24 cells; p = 0.2028 VAPB, p = 0.0316 VAPB at ERMCS; two-sided F-test, dFn=1 dFd=22, testing significance of the slope being different from zero).

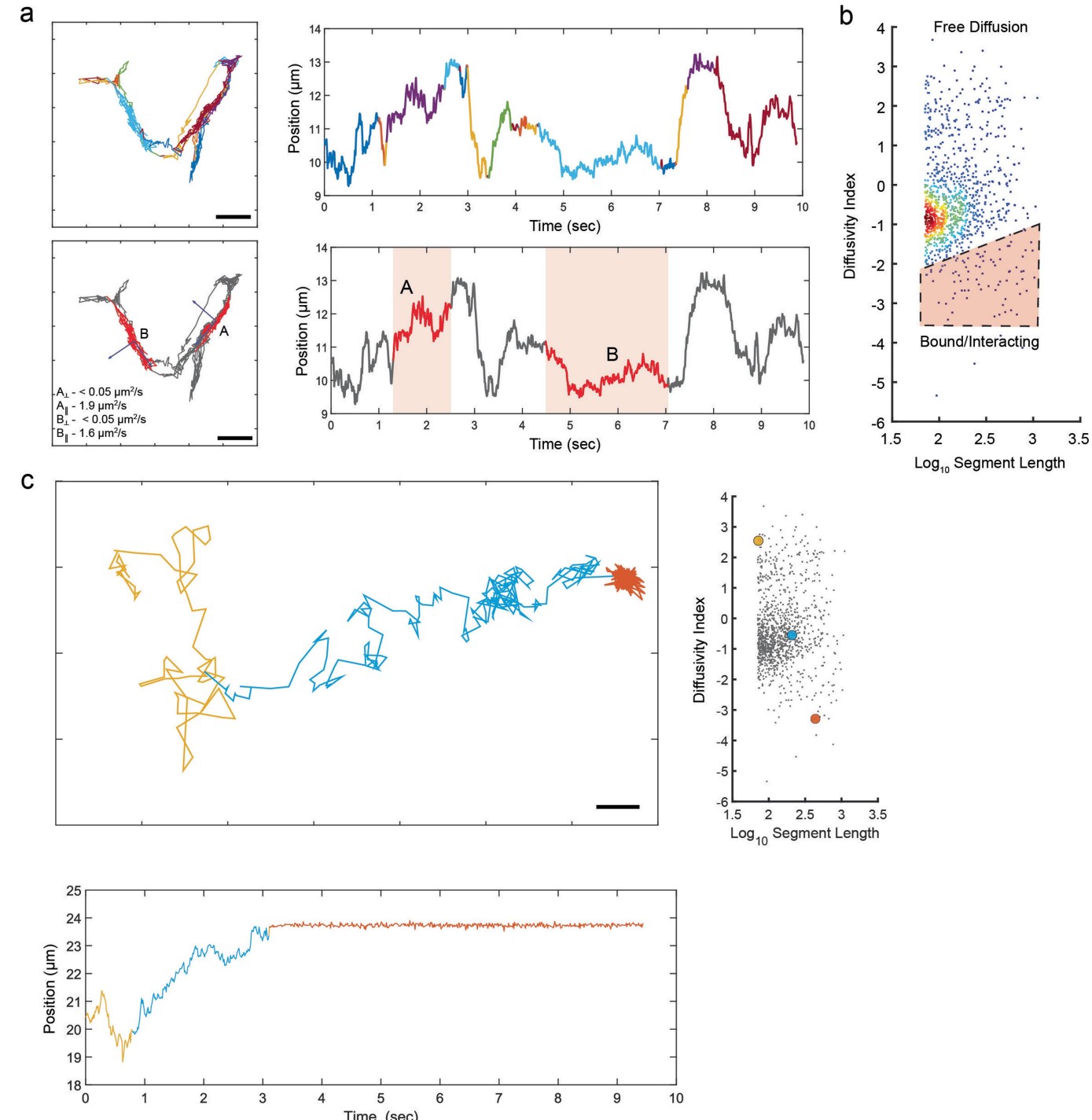

**Extended Data Fig. 4 | Latent states can be inferred from trajectory segments with similar mobility profiles. a**, Latent states determined in a single freely diffusing VAPB molecule within ER tubules (not shown). Two states with sufficient steps for analysis are highlighted in the lower panels, showing that diagonalization of the force and diffusion tensors reveals the native axis of the ER tubule (eiganvectors shown in blue). Diffusion in the perpendicular direction (around the tubule) is negligible, but diffusion along the long axis is discernible and consistent with literature values[68]. **b**, Segments of individual trajectories of VAPB can be characterized by the diffusivity index (see Supplementary Text), with bound states showing lower indices. A scatter plot of diffusivity index vs. segment length is shown, the red gate indicates segments considered to be interacting. **c**, A single VAPB trajectory with three discernible states. Segments of the trajectory defined to be in each state are colour coded in the xy-projection of the trajectory (upper) and in a time projection (lower). The diffusivity index for each trajectory is shown against the full set of analyzed VAPB trajectory segments in the scatter plot on the right, for context. Scale Bars: 500 nm.

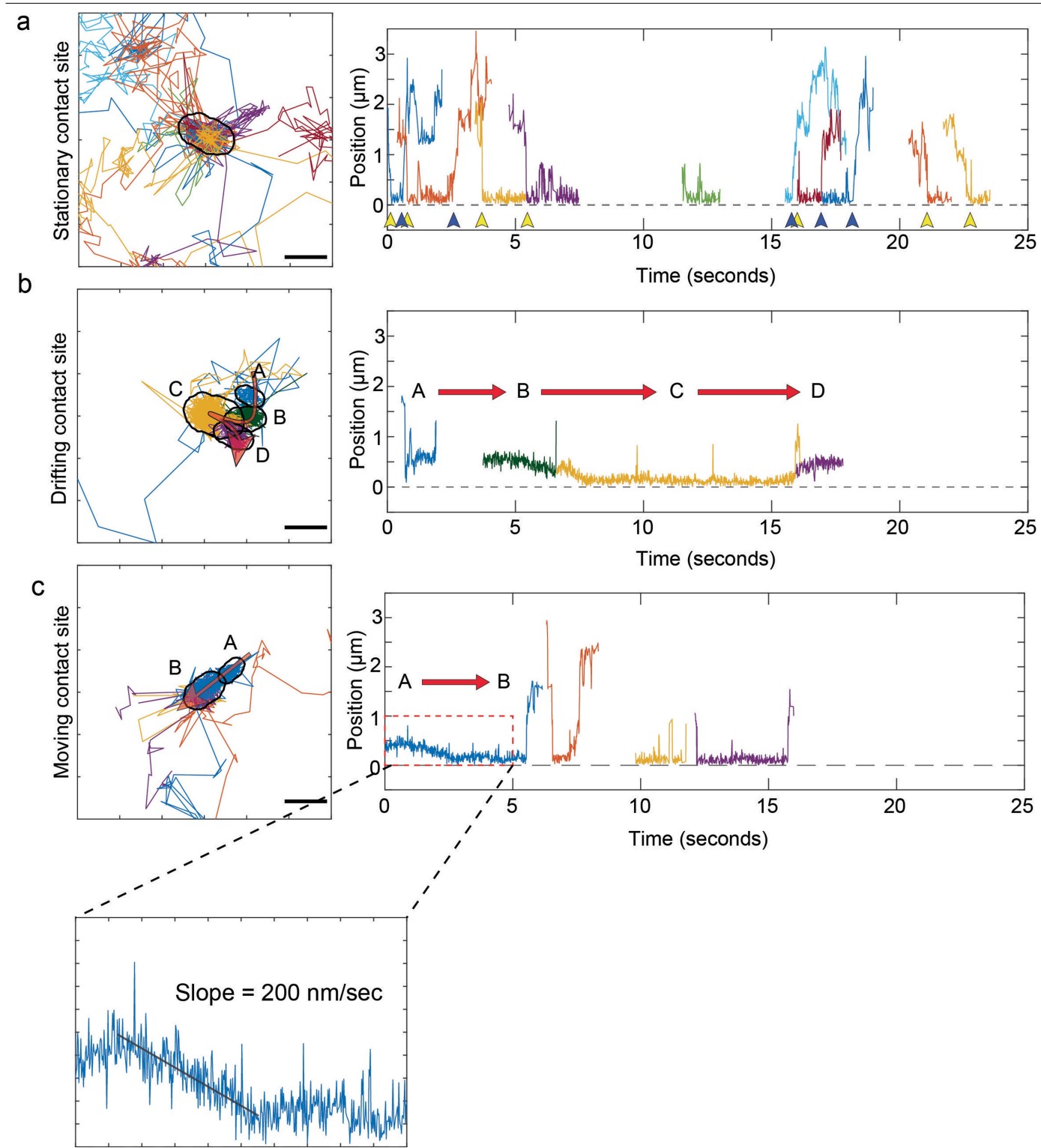

**Extended Data Fig. 5 | VAPB-mediated ERMCS contact site mobility is visible in VAPB trajectory analysis. a-c,** Representative examples of ER-mitochondria contact sites undergoing motion, showing a number of individual VAPB molecules exploring the site over the time scale shown. Panels to the left show trajectories colored by molecule, panels to the right show the distance to the center of the contact site over time for the trajectories on the left. Yellow arrows indicate entry events of a single VAPB molecule, blue arrows indicate exit events. Red arrows correspond to shifts in the mean location of confined trajectories over time, representing distinct contact site locations as a result of mitochondrial motion. Note the directed motion in the bottom example corresponds to a linear speed consistent with TAC-mediated motion in COS7 cells[69].

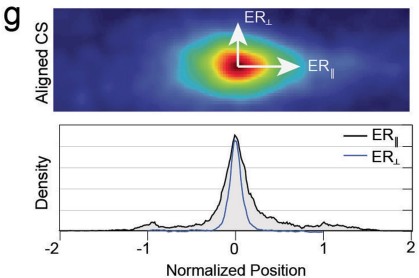

**Extended Data Fig. 6 | Stationary contact sites show spatially stable regions of common molecular behaviour. a**, A set of VAPB trajectories associated with a single ER-mitochondrial contact site, colored by trajectory. **b**, The xy-projections and time projections of trajectories of two distinct VAPB molecules that entered and left the contact site in a) are shown. Segments are color coded by their state, as defined in Extended Data Fig. 4b. Points where the molecules changed from freely diffusing states to confined states or vice versa are indicated with black dots and yellow arrows. **c**, A zoomed view of the contact site in a) scaled to match the subsequent panels. **d**, The probability density of finding a VAPB molecule across the contact site integrated over the 60 s time window it was deemed to be stationary. **e**, Voronoi tessellations derived from the density in d) used to segment individual steps of the VAPB trajectories for spatially-defined diffusion analysis. **f**, The effective 2D diffusion coefficient of the steps in e), grouped and solved by tessellation. Scale bars: a, 1 μm; b, 1 μm, insets, 200 nm; c-f, 200 nm.

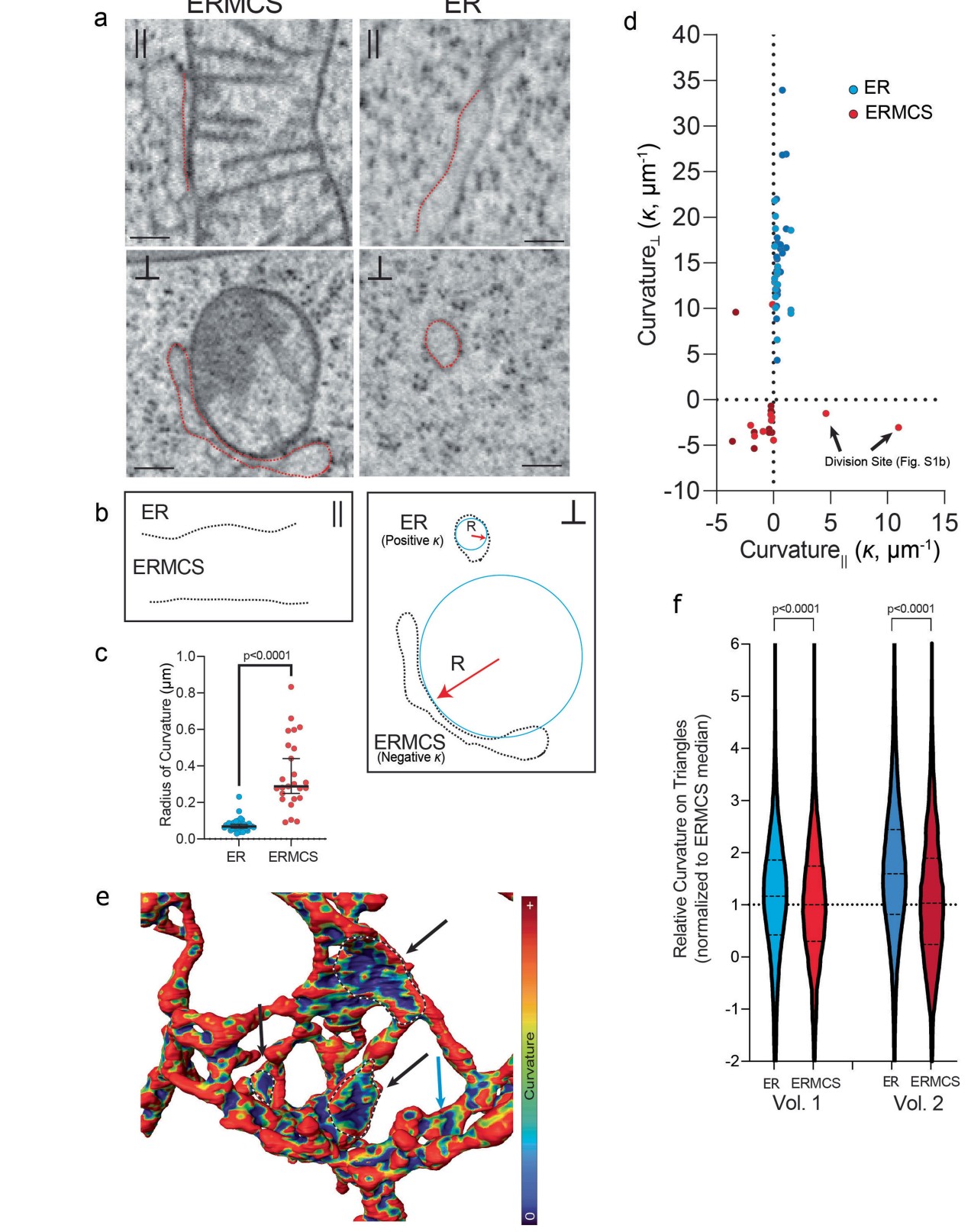

**Extended Data Fig. 7** | See next page for caption.

**Extended Data Fig. 7 | Changes in ER curvature within ERMCSs. a**, Single slices through the longitudinal (parallel to the central cylindrical axis of the structure) and perpendicular axes of a representative ERMCS or a neighboring ER control in the cytoplasm. Red lines indicate the location of the ER membrane. These are single representative images of n = 39 (ER tubules) or n = 25 (ERMCSs) collected through the two FIB-SEM volumes. **b**, Schematic of the ER cross sections in a and their methods of analysis. Note that in the longitudinal sections, ER membrane curvature ($\kappa$) is largely neutral, but in the perpendicular sections it can be highly positively curved (curving towards ER lumen) or negatively curved (curving away from ER lumen). The radius of curvature (R) is inversely proportional to the degree of curvature ($\kappa$). **c**, The radius of curvature of the ERMCS and isolated ER in the FIB-SEM volumes. Isolated ER controls are selected within 5 μm of each ERMCS (n = 39 tubules, 25 ERMCSs; p < 0.0001; two-sided Mann Whitney test). Error bars represent the median and 95% confidence interval for the median. **d**, Curvature along the parallel or perpendicular dimensions of the ERMCS or local ER controls as demonstrated in b. **e**, Local ER curvature as calculated in three dimensions on the triangulated mesh generated from the surface of the ER membrane in a representative FIB-SEM volume. Sites of contact with mitochondria (not shown for clarity) are traced with dotted lines and indicated with black arrows. The blue arrow indicates a site of direct contact with a mitochondrion facing away from the viewer. **f**, Relative mean local curvature of ERMCS- or non-CS-associated ER membrane as calculated from the triangulated surface in the two volumes shown in Extended Data Fig. 1 and normalized to median value in the ERMCSs. Dotted lines in the plots show the median and quartiles. (ERMCS data is n = 133,494 [vol1] or n = 37,062 [vol2] triangulated faces within ERMCSs, control ER is n = 500,000 triangulated faces selected randomly from the ER outside the ERMCSs in the matched volume. P-values are approximated from the z distribution of a Kruskal-Wallis test using two-sided Dunn's multiple comparisons test to adjust direct comparisons for multiplicity). Scale Bars: 200 nm.

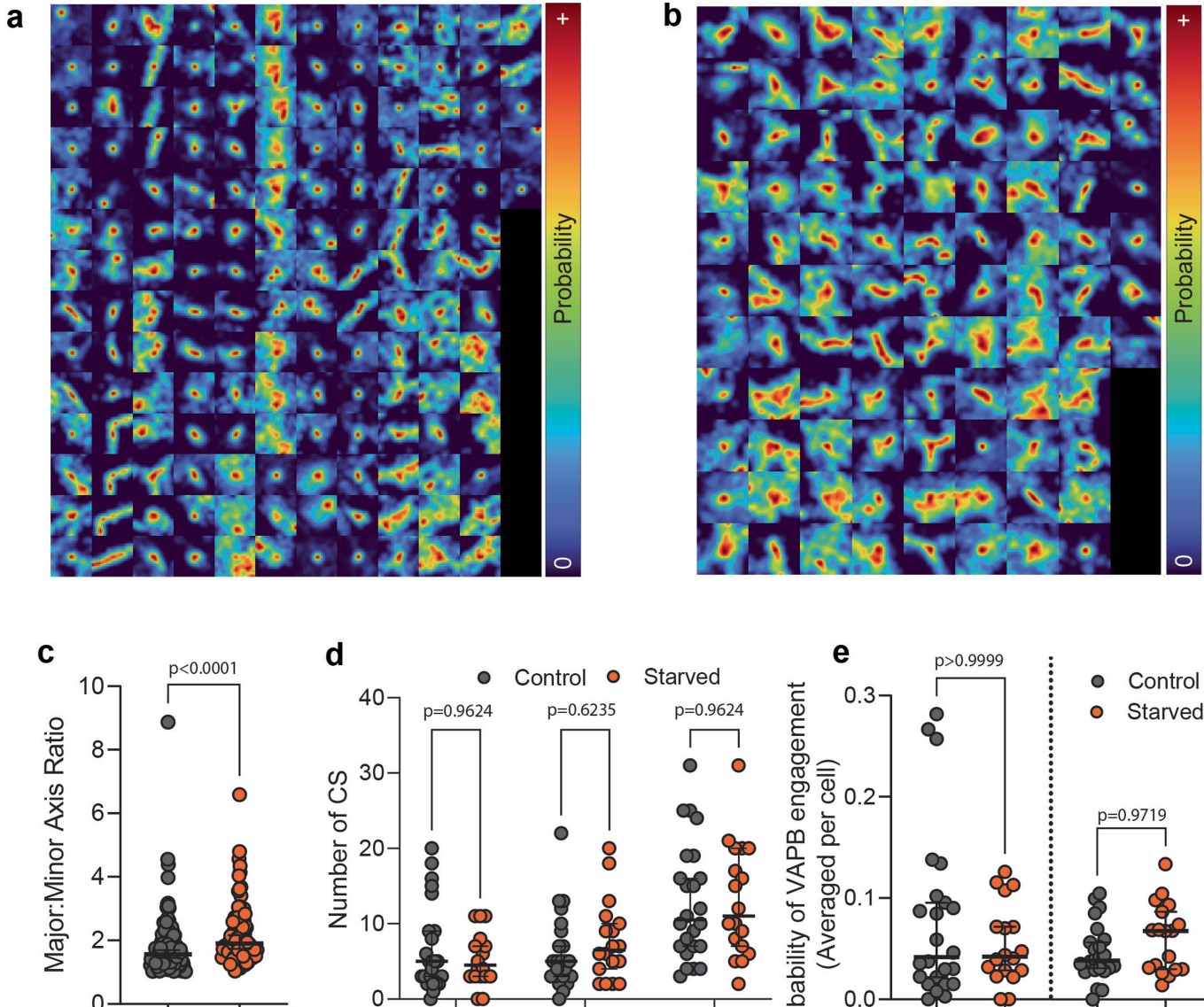

**Extended Data Fig. 8 | Variability of VAPB contact site size and shape in well fed and starved cells. a**, Spatially-defined probability of VAPB localization in 160 mitochondria-associated contact sites from cells cultured in complete DMEM. A 1.2 μm x 1.2 μm square around the center of each contact site is shown, note that proximal contact sites often appear in the images of their neighbors. **b**, 96 mitochondria-associated VAPB contact sites displayed as in a), for cells cultured in HBSS for 8 h. Note the asymmetric nature of contact site expansion under these conditions. **c**, The ratio of Major:Minor axis of ellipse fits to spatially-defined VAPB localization probability within individual contact sites, showing asymmetric expansion of contact sites under nutrient deprivation (n = 160 [Control], 96 [Starved]; p < 0.0001; Mann-Whitney test, two-sided). Error bars indicate the median and 95% confidence interval of the median. **d**, The number of mitochondria-associated (MitoCS), other

structure-associated (OtherCS), or total number of CSs in control cells or cells starved with an 8-hour incubation in HBSS (n = 24 [Control], n = 18 [Starved]; p = 0.9244 MitoCS, p = 0.2780 OtherCS, p = 0.8060 TotalCS; two-sided Mann-Whitney tests with Holm-Šidák threshold for multiple comparisons). Error bars indicate the median and 95% confidence interval of the median. No significant difference observed suggests continued regulation of VAPB at both classes of contact sites. **e**, The likelihood of VAPB engagement at ERMCSs or other contact sites in control cells or cells starved by 8 h in HBSS, averaged per each cell, showing VAPB is not largely depleted from the ER during starvation as in PTPIP51 overexpression (n = 24 DMEM, n = 18 HBSS; p > 0.9999 MitoCS, p = 0.9719 OtherCS, Dunn's multiple comparisons test, two-sided). Error bars indicate the median and 95% confidence interval of the median.

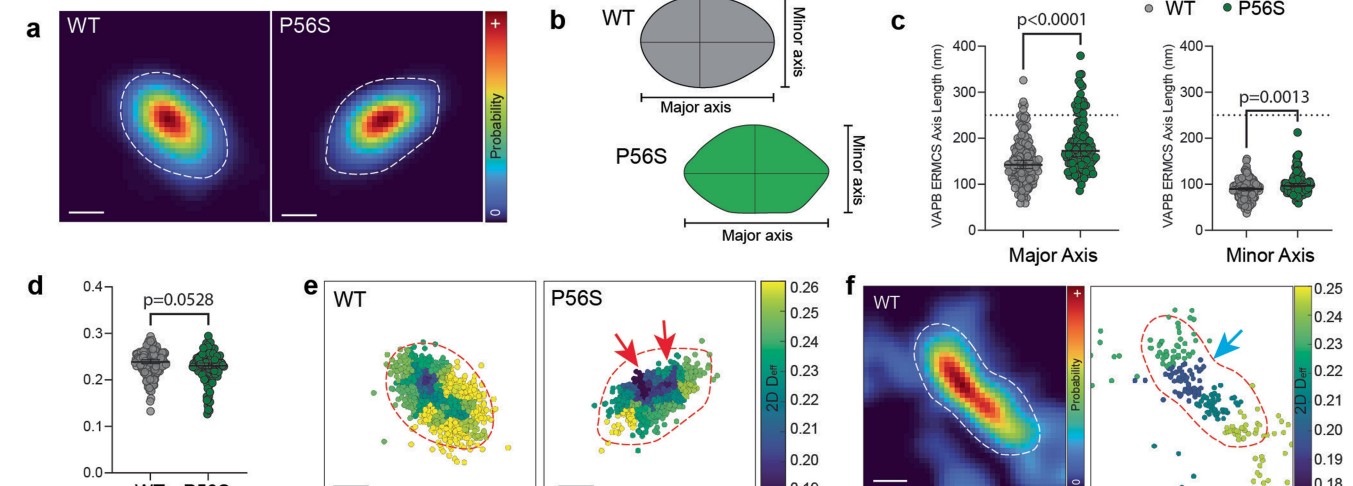

**Extended Data Fig. 9 | Additional characterization of ERMCS interactions of P56S VAPB molecules. a**, Spatially-defined probability of VAPB localization in representative WT or P56S VAPB-expressing cells. **b**, Schematic showing approximate major and minor axis of the examples ERMCSs in a. **c**, The size of the major and minor axes of ellipse fits to the probability of VAPB localization within the ERMCS, showing slight but significant expansion of ERMCSs in P56S VAPB-expressing cells (n = 160 [WT],118 [P56S]; p < 0.0001 Major, p = 0.0013 Minor; Dunn's multiple comparisons test, two-sided). Error bars indicate the

median and 95% confidence interval of the median. **d**, The 2D D$_{eff}$ of WT or P56S VAPB averaged over entire ERMCSs (n = 160 [WT], 118 [P56S]; p = 0.0528; Dunn's multiple comparisons test, two-sided). Error bars indicate the median and 95% confidence interval of the median. **e**, The diffusivity landscape of the WT or P56S VAPB ERMCSs shown in a. Red arrows indicate the asymmetric diffusion well at the center of the P56S VAPB-containing ERMCS. **f**, An example of a large WT VAPB-containing ERMCS similar in size to the one shown in Fig. 4d, showing a single, central diffusion well at the center of the contact site (blue arrow).

Jennifer Lippincott-Schwartz

# Reporting Summary

## Statistics

For all statistical analyses, confirm that the following items are present in the figure legend, table legend, main text, or Methods section.

| n/a | Confirmed | |
|---|---|---|
| ☐ | ☒ | The exact sample size (*n*) for each experimental group/condition, given as a discrete number and unit of measurement |
| ☐ | ☒ | A statement on whether measurements were taken from distinct samples or whether the same sample was measured repeatedly |
| ☐ | ☒ | The statistical test(s) used AND whether they are one- or two-sided<br>*Only common tests should be described solely by name; describe more complex techniques in the Methods section.* |
| ☐ | ☒ | A description of all covariates tested |
| ☐ | ☒ | A description of any assumptions or corrections, such as tests of normality and adjustment for multiple comparisons |
| ☐ | ☒ | A full description of the statistical parameters including central tendency (e.g. means) or other basic estimates (e.g. regression coefficient) AND variation (e.g. standard deviation) or associated estimates of uncertainty (e.g. confidence intervals) |
| ☐ | ☒ | For null hypothesis testing, the test statistic (e.g. *F*, *t*, *r*) with confidence intervals, effect sizes, degrees of freedom and *P* value noted<br>*Give P values as exact values whenever suitable.* |
| ☐ | ☒ | For Bayesian analysis, information on the choice of priors and Markov chain Monte Carlo settings |
| ☐ | ☒ | For hierarchical and complex designs, identification of the appropriate level for tests and full reporting of outcomes |
| ☒ | ☐ | Estimates of effect sizes (e.g. Cohen's *d*, Pearson's *r*), indicating how they were calculated |

*Our web collection on statistics for biologists contains articles on many of the points above.*

## Software and code

Policy information about availability of computer code

| Data collection | Nikon Elements AR 6.0, ZEN Black 2.3<br>The microscopes are operated using standard commercial software from Nikon (Nikon Elements) or Zeiss (ZEN Black), but can in principle be collected using any custom or public operation software for optical hardware. |
|---|---|
| Data analysis | Fiji-ImageJ (v.1.54f) with TrackMate plugin (v7.11.1)<br>Matlab 2022a (2015b for Ursa Analytics Code)<br>Python (v. 2.7.13)<br>All code used throughout manuscript is freely available at Github through the following repository (and associated linked repositories): https://github.com/cjobara/Obara-Nixon-Abell-2023-Nature |

For manuscripts utilizing custom algorithms or software that are central to the research but not yet described in published literature, software must be made available to editors and reviewers. We strongly encourage code deposition in a community repository (e.g. GitHub). See the Nature Portfolio guidelines for submitting code & software for further information.

## Data

Policy information about availability of data

All manuscripts must include a data availability statement. This statement should provide the following information, where applicable:
- Accession codes, unique identifiers, or web links for publicly available datasets
- A description of any restrictions on data availability
- For clinical datasets or third party data, please ensure that the statement adheres to our policy

The experimental data used in this paper are freely available in both raw and processed form at Figshare with the identifier: https://doi.org/10.25378/janelia.c.6916543.v1. All data are additionally available from the authors in other formats as needed upon request.

## Research involving human participants, their data, or biological material

Policy information about studies with human participants or human data. See also policy information about sex, gender (identity/presentation), and sexual orientation and race, ethnicity and racism.

| | |
|---|---|
| Reporting on sex and gender | N/A - There is no data of this type in the paper. |
| Reporting on race, ethnicity, or other socially relevant groupings | N/A - There is no data of this type in the paper. |
| Population characteristics | N/A - There is no data of this type in the paper. |
| Recruitment | N/A - There is no data of this type in the paper. |
| Ethics oversight | N/A - There is no data of this type in the paper. |

Note that full information on the approval of the study protocol must also be provided in the manuscript.

# Field-specific reporting

Please select the one below that is the best fit for your research. If you are not sure, read the appropriate sections before making your selection.

☒ Life sciences          ☐ Behavioural & social sciences          ☐ Ecological, evolutionary & environmental sciences

For a reference copy of the document with all sections, see nature.com/documents/nr-reporting-summary-flat.pdf

# Life sciences study design

All studies must disclose on these points even when the disclosure is negative.

| | |
|---|---|
| Sample size | Each SPT dataset contains a total of at least 16 regions (20.48 um x 20.48 um) each selected from a different cell divided over at least two experiments. Not all of these contained contact sites (see Extended Data Fig. 2). FIB-SEM data was visually examined in 3 COS7 cells and 1 U2-OS cell, but all data shown or quantified in the paper comes from a single representative COS7 cell. Single contact site analysis was performed on each of the hundreds of contact sites analyzed, each of which contains anywhere from 1-50 VAPB trajectories. No statistics were performed to determine appropriate sample size, and sample size per condition was largely governed by the technical limitations of the experiment and how many cells could be collected within the time window when the sample was healthy (see Supplemental Text, Sections IVA and VA for detailed explanation). All of the samples imaged showed variability across the entire range of possible values in the specific locations and functions of tethers, even over short time scales (see Ext. Data Fig. 2), and this was consistent in every independent experiment run. Thus, we feel the samples presented are sufficient to demonstrate the variation of the system, which is our major conclusion. |
| Data exclusions | No data was excluded from this study, though density-based identification of contact sites is performed manually based on the VAPB expression level in the cell. Thus, contact sites that do not have sufficient VAPB molecule interactions during the movie are likely removed from the dataset by lack of recognition. |
| Replication | All experiments in the paper were performed at least two independent times. No significant differences were observed between independent experiments--all samples examined showed consistent dynamic exchange and cell-to-cell variability across the range of potential tethering behaviors (see Extended Data Fig. 2 for quantification). The FIB-SEM volume presented contains two COS7 cells, and we visually checked the contact sites were similar in shape and abundance in both cells an additional two datasets that belong to other experimenters (one COS7 cell and one U2-OS cell). On account of the labor required, we only performed full reconstructions in the two subvolumes used in the paper, which come from the same cell (and a single control volume that is provided in the data repository but not the paper, since it did not have specific contact sites). All this data is provided in the Figshare repository. |
| Randomization | We did not randomize any aspects of the study, save the order in which datasets had manual thresholding applied. This was done in a random order to avoid bias about how thresholds were applied. |
| Blinding | SPT experiments are always performed essentially blind, since the experimenter cannot see the data while setting up the scope and initiating |

| Blinding | the imaging acquisition. Regions that are flat enough to perform SPT are selected using the structure of the ER (which is visible to the experimentalist). All downstream analysis is performed with the analyst blinded and the data in a scrambled order, thus, all priors and parameters are established blind. However, the majority of this pipeline is automated regardless, as such, there is little space for the analyst to affect the results even if they knew which dataset they worked with. |
|---|---|

# Reporting for specific materials, systems and methods

We require information from authors about some types of materials, experimental systems and methods used in many studies. Here, indicate whether each material, system or method listed is relevant to your study. If you are not sure if a list item applies to your research, read the appropriate section before selecting a response.

## Materials & experimental systems

| n/a | Involved in the study |
|---|---|
| ☒ | ☐ Antibodies |
| ☐ | ☒ Eukaryotic cell lines |
| ☒ | ☐ Palaeontology and archaeology |
| ☒ | ☐ Animals and other organisms |
| ☒ | ☐ Clinical data |
| ☒ | ☐ Dual use research of concern |
| ☒ | ☐ Plants |

## Methods

| n/a | Involved in the study |
|---|---|
| ☒ | ☐ ChIP-seq |
| ☒ | ☐ Flow cytometry |
| ☒ | ☐ MRI-based neuroimaging |

## Eukaryotic cell lines

Policy information about cell lines and Sex and Gender in Research

| Cell line source(s) | All experiments in the paper are performed using COS7 cells from ATCC, experiments are performed within 40 passages of the initial shock provided. |
|---|---|
| Authentication | We have not performed any authentication of the lines beyond purchasing a second aliquot from ATCC and performing an independent repeat of some of the SPT experiments in them. There were no obvious differences in any property examined. |
| Mycoplasma contamination | Cells were all free of mycoplasma at the time of experimentation, and are tested routinely during passaging. |
| Commonly misidentified lines (See ICLAC register) | No commonly misidentified cell lines were used in this study. |

