## [Peer Review File · Nature]

Manuscript Title: Motion of VAPB molecules reveals ER-mitochondria contact site subdomains

Reviewer Comments & Author Rebuttals

Reviewer Reports on the Initial Version:

Referees' comments:

Referee #1 (Remarks to the Author):

The authors have combined single particle tracking and fluorescence LM with FIB-SEM volumes to form a picture of the dynamics and structural characteristics of ERMCSs.

There are lots of interesting experiments (overexpression, diseased patient samples) that elucidate these “patches” on the ER and the change to their properties upon perturbation.

The experiments are mostly convincing and the work is overall excellent. Speculations and hypotheses, for instance the role of posttranslational modifications, are fun and intriguing.

Particularly informative is the remarkable discussion of Technical limitations at the beginning of the supplemental material. This is deeply appreciated.

However there were some pieces of analysis that I found lacking, and in some cases they are clearly required in order to be convinced that the conclusion is correct. These are detailed in the comments below.

Specific comments and questions:

1. Page 2: “regions of ER membrane within 24 nm of the outer mitochondria membrane were identified as sites of contact”. Please explain why 24 nm is chosen. Is there a general agreement in the community that 24 nm can be used to define as sites of contacts? References?
2. Page 3: “Simultaneous imaging of ER and mitochondria in conjunction with sptPALM (see Methods) confirmed that a majority of VAPB hotspots were located on regions of ER close to mitochondria (Fig. 1j, Video 2)”. Video 2 is not convincing that the vast majority of tracked molecules are associated with mitochondria. There are plenty of traces that are not coincident with mitochondria. The one on the top right seems to even avoid the mitochondrion as it goes by. Is this just this region? What is the distribution of traces that do contact mito vs those that don't? In this video it looks like less than 50% of traces are superimposed over mitochondria.
3. Page 4: “ER membranes dramatically deform to match local mitochondrial curvature.” How is it dramatic? Compared to what? Would it help to have numbers, such as for change of membrane curvature over a set distance compared to a control area? It is hard in Figure 2i to understand what region specifically the arrows are pointing to, and what their color is (curvature).
4. Page 5: “higher abundance of tethers towards the center of the contact site would be predicted to provide increased adhesive force between the two membranes, a phenomenon we could directly observe as regions of negative curvature in the center of ERMCSs in FIB-SEM datasets”. In Fig 2i, it looks like the arrows are pointing to regions of positive curvature, not negative. Secondly please expand on why negative curvature is associated with “adhesiveness”. Is there an actual mechanistic definition of “adhesiveness” with respect to membrane curvature? If yes, please give the reference.

Referee #2 (Remarks to the Author):

Obara et al use a combination of 3D electron microscopy and single particle tracking to interrogate the structure and dynamics of VAP-mediated ER-mitochondria contact sites (ERMCS). They describe rapid turnover of tethers at the contacts, despite the overall stability of the contacts. They further reveal sub-compartmental microdomains and a change in contact site morphology depending on the cellular state. This study represents a very elegant survey of ERMCS and involves substantial technical advances. The data and images generated are beautiful and the techniques are solid. The manuscript sets a very high standard for studies of membrane contact sites. Advance of knowledge, however, is limited. Several of the findings reported here for ERMCS have already been described in the literature for other VAP-mediated contact sites, particularly ER/plasma membrane (ER/PM) junctions, although not with the same level of quality.

Specific Comments

- Statement at the beginning of the Discussion “Existing models of ERMCSs generally depict stable protein complexes tethering ER and mitochondria together”. The basis for this statement is not clear. Based on studies of other contact sites it is already implicit that these are highly dynamic microdomains with a high degree of constant protein turnover.
- The cartoon illustration in Fig 3a suggests that binding occurs between the VAP MSP domain and the PTPIP51 TPR domain which is incorrect. Binding occurs between the VAP MSP and the PTPIP51 FFAT motif (aa 157-172)
- VAP is known to dimerize and no mention of this is made within the text. This will impact the findings, particularly with respect to the P56S mutation, a dominant mutation. Since VAP dimerizes, the exogenously expressed P56S mutants may form partially functional dimers, thus impacting motility. The authors should speculate on the role of dimerization.
- The single molecule trajectories in figure 1e (iii & iv) quite clearly show a trajectory that is confined, but the trajectories are non-overlapping as though this contact site was ‘drifting,’ as described in the extended data. Why was this figure chosen if analysis was performed on stationary contacts?
- The present focus on ERMCSs is novel and important, but these findings would be better appreciated if placed within the greater context of other VAP-mediated contact sites, and more generally other contact sites. ‘Donut-like’ subdomains have been reported for proteins inhabiting ER/PM contact sites with residents either inhabiting the periphery or the center of the contact site akin to what the authors hypothesize for ERMCS. ER/PM junctions are known to change size in response to various stimuli and cellular state, including those mediated by VAP (PMID: 26028218, PMID: 30012696). Single particle tracking of VAP partners at the ER/PM junction has shown them to be fully mobile but confined to within the perimeter of the contact site (PMID: 21464280). VAP

proteins at ER/PM junctions have been previously hypothesized to bind, unbind, and rebind partners (it was proposed that this may be how phosphatases access phospho-FFAT motifs on VAP partners [PMID: 29941597]). Single particle tracking of STIM1/Orai has shown their interaction to be surprisingly dynamic with constant diffusional turnover suggesting that this is not just VAP-specific but is likely a feature of proteins that tether any two membranes together (PMID: 25057023). Discussion of the many similarities (and any differences found) between the properties of the ERMCS and the ER/PM contact site (and other contact sites) would be beneficial. Especially since VAP proteins inhabit and help to generate both spaces.

- What percentage of your tracks exhibit a change in confined to freely diffusive behavior or the opposite when entering or leaving a presumed contact site? Did this change with respect to expression level of the VAP construct? If mitochondrial partners are the limiting factor in VAP localization to the ERMCS, then overexpression of exogenous VAP will saturate the endogenous binding partners, the bulk of the tracked VAP proteins will be less likely to interact, and the average diffusion coefficients will be artificially altered.

Minor Comments

- In Extended Data Figure 4, 'are indicated with black dots and blue arrows' should be changed to '...black dots and yellow arrows.'
- There was no literature list attached for the main text in the material I received. I received a literature list but only for the methods section and the expanded data section. I was able to find the missing literature section in the bioRxiv version of this paper, however.

Referee #3 (Remarks to the Author):

The manuscript by Obara et al. visualized the transient, small and sensitive dynamic molecular bonds within mito-ER contact sites by using sptPALM. They also characterize ER-mito contact site (ERMCS) morphology using FIB-SEM imaging. They show that the VAPB molecule, which is the main ER-resident tether, is dynamic but shows enhanced enrichment and reduced diffusivity in zones that correlate with ER-mito proximity. They use the VAPB enrichment to define the contact site and show that the size of the contact site was sensitive to the abundance of the mito tethering molecule PITPIP51 but not to the abundance of VAPB. They also show that contact sites expand under starvation conditions, but the VAPB molecules remain dynamic. Finally, they examine ALS mutants of VAPB and show that they have subtle changes in their contact site diffusivity, becoming trapped in subdomains.

Overall, the manuscript is a tour de force imaging achievement. The visualization of VAPB dynamics together with markers of ER and mitochondria is highly technically challenging and the authors carried out a lot of optimizations for the data acquisition to achieve this task. In addition, they did a careful job with the analysis of the sptPALM data, considering several complications such as the

impact of the ER geometry on the molecular motion and using appropriate analysis algorithms to extract the dynamic parameters. The FIB-SEM images are also impressive. All that being said, the methodologies are not really new and it is not clear that a lot of new biological insight has been gained from the technical tour de force. The main conclusion is that the tethering molecules are dynamic rather than stationary. This is not too surprising given what we know about many other similar tethering or adhesion molecules. For example, molecular machinery within focal adhesions is highly dynamic, the synaptic adhesion molecules (neuroligin and neuroligin) are highly dynamic. Hence, the model of maintaining an adhesive contact structure through dynamic interactions of adhesion molecules is not new. In fact, it would have been much more surprising if VAPB was stationary at the contact site. In addition, the part about the role of VAPB mutants in ALS is highly speculative. The authors show some subtle changes to the mobility of VAPB at contact sites but the mechanism for these changes and the physiological relevance to dysfunction in ALS are not explored. The work remains descriptive and for a high impact study, it would have been worth exploring how perturbing the tethering dynamics or the area of contact site impact ERMCS function and cell physiology.

Below are more specific comments for the authors:

Major comments:

I am a bit confused about how contact sites are identified. In the manuscript the authors comment that “we looked for locations where ER-localized tethers exhibited patterns of reduced single molecule motion, reasoning these may reveal the sites where they interact with their mitochondrial binding partners.” They then quantify the diffusivity of VAPB and show that it is reduced at contact sites. This seems like a circular logic if they are identifying contact sites as those where VAPB is less mobile. I may be confused about this, but it needs some clarification.

“These probability “hot spots” were present in the VAPB dataset but not with Halo-TA (Fig. 1f), and they were consistent with the size of ERMCSs extracted from FIB-SEM volumes (Fig. 1g-i).” This sentence is misleading because reading more deeply into the figure caption and SI, it seems that the authors measured contact site in FIB-SEM indirectly by simulating single molecule trajectories projected onto the FIB-SEM images and what they are comparing is the contact site size measured from these simulated traces, not directly from the FIB-SEM images. It would be informative to also include the projected size of the contact site directly measured from the FIB-SEM images so that the reader can have an idea about how these two methods compare. In addition, I suggest rewording this part of the text to be more transparent about the simulation.

The statistics are poorly reported throughout the manuscript. I could not find the number of cells or the number of biological replicates or the type of statistical test performed for the different quantifications. I suggest the authors add this information in the figure captions. In addition, it seems the statistical analysis is done by considering each trace or contact site as an independent data point rather than comparing the means of biological replicates. Subtle differences will seem artificially statistically significant simply due the large number of data points when a trace or contact site is considered for the statistical analysis. I suggest the authors use super plots and report statistics on biological replicates.

Figure 4h-j, the observation that the P56S mutant explored only subdomain of the contact site should be supported by a quantification of area of confinement in comparison to the overall contact site area. This part of the manuscript is mainly supported by anecdotal examples. From the examples shown, it is not clear how the mutant is different from the WT and what exactly the reader should be seeing in the color-coded segmentations. Is there a hypothesis as to why the P56S only surveys subdomains if it is not aggregated in the contact site? The mechanism or the physiological relevance of this observation are unclear.

Minor comments:

Can author specify why distance between mitochondria and ER less than 24 nm was chosen as a cutoff for defining contact sites?

“The affinity of VAPB for FFAT-containing binding partners like PTPIP51 is very low^{21,47,48}, suggesting contact site interaction may consist of very many rapid binding and unbinding events across the structure”. The authors may consider extracting off rates to compare to the reported low affinity, and if such affinity (kd) is known, kon can be also estimated, which would provide a more quantitative view of VAPB dynamics on contact sites.

Referee #4 (Remarks to the Author):

The manuscript presents data on the movement of the mitochondria-ER contact (MERC) tethering protein VAPB. Through single-molecule tracking, the authors show fascinating data that demonstrate that a fusion protein based on this tether occasionally shows targeted overlap of its signal with MERCs. This is not seen with a minimally tail-anchored protein. The strength of the manuscript lies in the single-molecule tracking of VAPB that provides important and exciting insight about its extent of movement within and outside of MERCs. This makes Video 3 a true highlight of this study and membrane contact site (MCS) research in general.

Unfortunately, while I am impressed with this aspect of the study and also enthusiastic about any improvements that the authors will do, the study in its current form omits several aspects of what is known about MERC biology, for instance the distinction of rough and smooth MERCs or the role of autophagy for MERCs. This is reflected in the omission of key studies in the reference list, which also appears not matched to the text. The authors touch upon MERC biology by investigating starvation-relevance of their observation. However, this aspect is currently much underdeveloped, relies on a very long incubation period in starvation medium that is known to cause autophagy and does not address other, better characterized mechanisms that could give cleaner readouts. The study is also relying exclusively on one type of MERC tether, but many more are known, which reduces the overall relevance for the field. Another attempt to introduce significance into their findings uses a known ALS mutation of VAPB. While this dataset is strong, it is not mechanistically explained and could suffer from the known role of this mutant to alter overall ER structure and lipid composition. Lastly, another major weakness of the study is that no knockouts or knockdowns of well-known factors for MERC formation have been put into context of the VAPB behavior. Together, these

deficiencies currently do not allow much mechanistic insight.

Overall, very strong descriptive data on VAPB are in the manuscript that is a candidate for nature. Unfortunately, the manuscript currently is underdeveloped, and contains no experiments that provide a minimum of information about the universality of the observations. Once at least a few of such experiments are provided, the study would match the claims made in the title.

Specific points

1. The field of MERCs has recently seen a critical refinement, where distinctions between rough MERCs and smooth MERCs have been distinguished (see Anastasia et al. Cell Reports 2021). Unfortunately, the authors do not provide any information about this property of the structures they describe in their initial characterization of MERCs. Accordingly, they describe their data as “this view of ERMCS ultrastructure”, but due to the aforementioned point, this is therefore an overstatement.

2. While I have no objection to the use of VAPB, since this is an important MERC tether, the manuscript falls short in terms of an investigation of its relative importance. The authors initially state that it “likely” acts as “a tether for other organelles”. This is a rather casual observation and decreases the otherwise quantitative aspects of this study. At a minimum, the authors should give an idea about the frequency of MERC localization of VAPB versus other enrichments early on. This is provided in part by Figure 3F. However, this figure lacks a calculation of the relative abundance of MERC-localization of VAPB versus other contact sites. Also, I am not sure about the significance of the presented data. While it is true that in the presence of high amounts of PTP51, the number of non-mito MCS decreases, it is unclear whether the data presented by the authors actually show a specific reduction of the relative amounts of VAPB on non-mitochondrial contact sites. Could the authors produce a “coefficient” that measures relative amounts on both types of MCS? Moreover, it would be better if they actually identify the nature of those enrichments that do not contact mitochondria.

3. The extent of ER membrane curvature is correlated to distinct lipid species. For instance, PE could mediate increased curvature and microscopic techniques exist to localize membrane tension developed by the Matile and Roux labs (e.g., Colom et al., Nature Chemistry 2018).

4. Many physiological conditions exist, which increase MERCs. A key mechanism identified by the Hajnoczky and Lavandro labs is short-term ER stress. I am surprised that the authors do not investigate this mechanism. Similarly, the interference with curvature-inducing PE using, for instance cinnamycin or a PSD1 mutation could yield more relevant findings. Instead, they focus on nutrient deprivation. This condition has the disadvantage of also leading to autophagy, for instance the authors should have a look at the study by Hamasaki et al., nature 2013. This choice therefore requires investigation of autophagosome localization relative to MERCs and their amounts. Another issue is that during a time course of starvation, more or less MERCs could appear due to this issue. A control of the extent of MERC formation during the very long 8h timeperiod chosen by the authors should control for this possibility. Moreover, they again do not consider the data recently reported that distinguish between rough and smooth MERCs.

5. While the data on the VAPB mutant are strong, they are not mechanistically explained. How does

this mutant achieve this change in motility? It is likely misfolded, so are we just looking at the consequence of this? More questions arise when considering that this mutant disrupts normal ER morphology, thus suggesting that the observations are a side-effect of this property.

6. Overall, the decision to exclusively focus on the VAPB tether limit the interest in this study right now. While many presented datasets are very strong, they are restricted to this tether and do not provide general information about tethers or MERCs. This limitation is puzzling, given that many more tethering proteins are known, in addition to many other factors that are required for MERC formation, and at least a couple should be analyzed in a similar manner to give readers information that is not restricted to VAPB. This is particularly problematic, given the title promises much more than just a study about VAPB. At an absolute minimum, the authors must be more honest with the title and clearly declare that the manuscript deals with one tether type, the VAPB-PTPIP51 pair. Such a decision would, however, reduce the enthusiasm that I otherwise have for the experiments presented in manuscript.

7. Previous studies on MERCs in nature have used knockouts or knockdowns for key MERC regulatory proteins (e.g., de Brito et al. 2008, Hamasaki et al., 2013). The current study contains none, thus seriously limiting the scope of the study.

Minor points:

1. The authors chose to use a highly uncommon abbreviation for mitochondria-ER contacts. The field has homed in on the terms MAMs for biochemical isolates of these contacts and MERCs for the physical contact sites. The vast majority of the existing literature uses these terms. It is not a good idea to keep changing terms for this structure and the authors must use the consensus abbreviation, especially in an important journal such as nature.

2. It looks like the reference numbering is wrong. For instance, the authors refer to #21 and #44 for the ALS mutants of VAPB but those references do not deal with this question. In fact, from the list provided by the authors, I am unable to identify the papers where this had been identified.

Author Rebuttals to Initial Comments:

Response to Referees' comments:

Referee #1:

The authors have combined single particle tracking and fluorescence LM with FIB-SEM volumes to form a picture of the dynamics and structural characteristics of ERMCSs. There are lots of interesting experiments (overexpression, diseased patient samples) that elucidate these “patches” on the ER and the change to their properties upon perturbation. The experiments are mostly convincing and the work is overall excellent. Speculations and hypotheses, for instance the role of posttranslational modifications, are fun and intriguing. Particularly informative is the remarkable discussion of Technical limitations at the beginning of the supplemental material. This is deeply appreciated. However, there were some pieces of analysis that I found lacking, and in some cases they are clearly required in order to be convinced that the conclusion is correct. These are detailed in the comments below.

Specific comments and questions:

1. Page 2: “regions of ER membrane within 24 nm of the outer mitochondria membrane were identified as sites of contact”. Please explain why 24 nm is chosen. Is there a general agreement in the community that 24 nm can be used to define as sites of contacts? References?

We thank the reviewer for this comment. The literature is inconsistent regarding intermembrane spacing for one “type” of contact versus another (reviewed in PMID: 27341186). For this study, we focus on contact sites formed by VAPB and PTPIP51. Studies measuring the intermembrane space for overexpressed VAPB and PTPIP51 suggest it is in the range of 20 nm (PMID: 24893131), and AlphaFold structures for the pair loosely agree with this distance (with a maximum stretching distance < 30 nm). We selected 24 nm instead of 20 nm as the upper limit for contact site distance because our FIB-SEM data was collected using 8nm voxels, and as such, multiples of 8 nm minimize the contribution of smoothing artifacts. We have rewritten the main text to explicitly state our reasoning for this threshold, and we have added Supplementary Text (Sections 2a and 3b) to state how this was achieved and the implications.

2. Page 3: “Simultaneous imaging of ER and mitochondria in conjunction with sptPALM (see Methods) confirmed that a majority of VAPB hotspots were located on regions of ER close to mitochondria (Fig. 1j, Video 2)”. Video 2 is not convincing that the vast majority of tracked molecules are associated with mitochondria. There are plenty of traces that are not coincident with mitochondria. The one on the top right seems to even avoid the mitochondrion as it goes by. Is this just this region? What is the distribution of traces that do contact mito vs those that don't? In this video it looks like less than 50% of traces are superimposed over mitochondria.

The reviewer is correct in saying that the majority of VAPB trajectories (~80-90%) are freely diffusing in the ER and not anchored to mitochondria. Indeed, this is why ERMCSs are so hard to see by diffraction limited imaging. What we are highlighting in Fig. 1j and Video 2 are sites of increased VAPB localization density, where VAPB trajectories appear to be interacting at a specific site. These sites or 'hotspots' are often (but not always) preferentially next to mitochondria and are not easily seen by looking only at raw trajectories, as shown in Movie/Video 2 to illustrate the method. To address the reviewer's concern, we have added Extended Data Figs. 2 + 3, where we directly quantify the frequency of VAPB interactions with mitochondria through several metrics. We have further updated the main text and added Supplementary Text (Sections 7a-d) to demonstrate the method by which this was achieved. While most cells show less than 10% of their VAPB interacting with mitochondria at any given moment, we note that those same cells often show 50% or more of their VAPB hotspots associated with mitochondria. Thus, a small fraction of the total population of VAPB molecules can be responsible for the important biological interactions, an observation further underscored by our results with the disease-causing P56S mutation.

3. Page 4: "ER membranes dramatically deform to match local mitochondrial curvature." How is it dramatic? Compared to what? Would it help to have numbers, such as for change of membrane curvature over a set distance compared to a control area? It is hard in Figure 2i to understand what region specifically the arrows are pointing to, and what their color is (curvature).

We thank the reviewer for this excellent suggestion. We have developed a new curvature analysis approach to measure the extent of change in ER curvature across the entire contact site (global curvature), which can be measured in units that are much more biologically relevant (radius of curvature in microns and degree of positive or negative curvature) in Extended Data Fig. 7. We have detailed both the original local curvature method and the new global curvature method in Supplementary Text (Sections 3d-e). We have also fixed the arrows in figure 2i (now figure 2l).

4. Page 5: "higher abundance of tethers towards the center of the contact site would be predicted to provide increased adhesive force between the two membranes, a phenomenon we could directly observe as regions of negative curvature in the center of ERMCSs in FIB-SEM datasets". In Fig 2i, it looks like the arrows are pointing to regions of positive curvature, not negative.

We apologize for this mistake and have corrected the positions of the arrows.

4b. Secondly please expand on why negative curvature is associated with "adhesiveness". Is there an actual mechanistic definition of "adhesiveness" with respect to membrane curvature? If yes, please give the reference.

This is a good point. We are unaware of any direct measurements of endogenous ER rigidity, which would allow an estimation of the adhesive forces required to deform the membrane by a specific amount. Our point here is to note that the only place in our FIB-SEM volumes that we see ER membrane showing an extended region of negative curvature is where ER is directly contacting mitochondria. This is consistent with a bending force toward mitochondria pulling the ER membrane away from its normal, positively curved, tubular shape. To help clarify this point, we have added Extended Data Fig. 7, characterizing the extent and direction of curvature in ERMCSs, as well as location-matched ER controls (i.e., ER not contacting any other organelles). Indeed, the radius of curvature in ERMCSs is consistent with the radius of mitochondria in this dataset, and the curvature is folded away from the ER lumen (negative curvature), in contrast to the ER controls which are universally positive and show much smaller radii of curvature (as expected for tubular ER in the periphery).

Referee #2:

Obara et al use a combination of 3D electron microscopy and single particle tracking to interrogate the structure and dynamics of VAP-mediated ER-mitochondria contact sites (ERMCS). They describe rapid turnover of tethers at the contacts, despite the overall stability of the contacts. They further reveal sub-compartmental microdomains and a change in contact site morphology depending on the cellular state. This study represents a very elegant survey of ERMCS and involves substantial technical advances. The data and images generated are beautiful and the techniques are solid. The manuscript sets a very high standard for studies of membrane contact sites. Advance of knowledge, however, is limited. Several of the findings reported here for ERMCS have already been described in the literature for other VAP-mediated contact sites, particularly ER/plasma membrane (ER/PM) junctions, although not with the same level of quality.

Specific Comments

1. Statement at the beginning of the Discussion “Existing models of ERMCSs generally depict stable protein complexes tethering ER and mitochondria together”. The basis for this statement is not clear. Based on studies of other contact sites it is already implicit that these are highly dynamic microdomains with a high degree of constant protein turnover.

We thank the reviewer for bringing up this point. We agree that prior ER/PM contact site studies have suggested these sites are microdomains with constant protein turnover. However, this dynamism occurs on a completely different timescale than what we demonstrate with ERMCSs. Exchange of molecular tethers in and out of ER-PM contact sites has been reported to occur on the order of tens of seconds to minutes based on photoactivation measurements (e.g., PMID: 26028218). ERMCSs have proved refractory to this experimental approach, suggesting much faster interactions, which we could directly observe using our sptPALM approach. To further demonstrate the significant dynamics of VAPB at ERMCSs, we have added new data directly quantifying the binding kinetics of

VAPB at ERMCSs (Fig. 2a-d), finding the median dwell time of molecules to be in the range of 500 msec (at least 10-fold more transient than VAP-A at the PM). This new information is now presented and discussed in the main text. We have also included new Supplementary Text (Sections 2b and 2c) to discuss the extensive literature in this field and provide context to the exciting nature of our results.

2. The cartoon illustration in Fig 3a suggests that binding occurs between the VAP MSP domain and the PTPIP51 TPR domain which is incorrect. Binding occurs between the VAP MSP and the PTPIP51 FFAT motif (aa 157-172).

We thank the reviewer for catching this error. We have updated the cartoon appropriately.

3. VAP is known to dimerize and no mention of this is made within the text. This will impact the findings, particularly with respect to the P56S mutation, a dominant mutation. Since VAP dimerizes, the exogenously expressed P56S mutants may form partially functional dimers, thus impacting motility. The authors should speculate on the role of dimerization.

We thank the reviewer for raising this issue. We now speculate on this topic in the discussion by stating that the “inability of these trapped molecules (i.e., P56S VAPB) to leave the ERMCS may lead an impaired ability of the contact site to undergo normal dynamic restructuring, either through altered PTPIP51 interactions or by changes in dimerization/lateral aggregation of VAPB within the contact site”. We further provide a more expanded discussion of VAPB dimerization in the Supplement, including its potential effects on the behavior of the VAP-P56S mutant (Supplementary Text, Section 2d).

4. The single molecule trajectories in figure 1e (iii & iv) quite clearly show a trajectory that is confined, but the trajectories are non-overlapping as though this contact site was ‘drifting,’ as described in the extended data. Why was this figure chosen if analysis was performed on stationary contacts?

We agree with the reviewer that there is drifting in some of the contact sites in this figure. ER and mitochondria both can move within the cytoplasm, so ERMCSs can move or drift with them. The majority of our analysis is insensitive to contact site motion (probabilistic analysis, single trajectory analysis, etc.), and so for these approaches all contact sites are retained. We exclude moving contact sites only from our neighborhood-based approach (Extended Data Fig. 6 and Figs. 2j-k, 3k-l, 4d) since drifting is a problem in this type of analysis. We provide an additional discussion of these different approaches to ERMCS analysis in Supplementary Text (Sections 6a-e and 7a-d). We have also added the full sample sizes and statistics to the figure captions.

5. The present focus on ERMCSs is novel and important, but these findings would be better

appreciated if placed within the greater context of other VAP-mediated contact sites, and more generally other contact sites. 'Donut-like' subdomains have been reported for proteins inhabiting ER/PM contact sites with residents either inhabiting the periphery or the center of the contact site akin to what the authors hypothesize for ERMCS. ER/PM junctions are known to change size in response to various stimuli and cellular state, including those mediated by VAP (PMID: 26028218, PMID: 30012696). Single particle tracking of VAP partners at the ER/PM junction has shown them to be fully mobile but confined to within the perimeter of the contact site (PMID: 21464280). VAP proteins at ER/PM junctions have been previously hypothesized to bind, unbind, and rebind partners (it was proposed that this may be how phosphatases access phospho-FFAT motifs on VAP partners [PMID: 29941597]). Single particle tracking of STIM1/Orai has shown their interaction to be surprisingly dynamic with constant diffusional turnover suggesting that this is not just VAP-specific but is likely a feature of proteins that tether any two membranes together (PMID: 25057023). Discussion of the many similarities (and any differences found) between the properties of the ERMCS and the ER/PM contact site (and other contact sites) would be beneficial. Especially since VAP proteins inhabit and help to generate both spaces.

We thank the reviewer for raising these points. We agree that one of the exciting aspects of these data is how they relate to ER-PM contact sites, and the remarkable differences in dynamics between the two types of structures. We have now revised both our result and discussion sections to reference the literature on ER-PM contact site dynamism. Since space limitations prevent a full detailing of this impressive body of literature, we also have added a supplementary section (Supplementary Text, Section 2d) that fully discusses ER-PM contact site literature in relation to our findings on ER-mitochondria contact sites. We should add that the techniques referenced by the reviewer are not possible in our system (i.e.--quantum dots are too large to track organelle-localized proteins non perturbatively, (PMID: 29941597), and ERMCSs tether exchange is too rapid for simultaneous GFP and mCherry burst movies (PMID: 25057023)). However, our HaloTag-based approach should work quite well at ER-PM junctions, so future work will allow a direct comparison of the dynamics at these different types of contact sites with the same tool.

6. What percentage of your tracks exhibit a change in confined to freely diffusive behavior or the opposite when entering or leaving a presumed contact site?

We thank the reviewer for this point. We have now explicitly quantified each of these variables (Extended Data Fig. 3). Briefly, all interacting molecules in ERMCSs show a shift to freely diffusing behavior when they leave the contact site. Determining whether all freely moving molecules entering a contact site shift to confined motion is more difficult to assess since some of these molecules can move along ER facing on the opposite side of the contact site and never slow down or enter the contact site subdomain since they do not interact with mitochondrial tethers. The average likelihood of interaction (when grouped by cell) was around 50%, but there was significant heterogeneity in this number (some cells have more contact sites that seem to be growing during the acquisition (i.e., number of

binding events>>number of leaving events) and others have more that seem to be shrinking). A thorough quantitative analysis of this will require more long-lasting acquisitions and thus is for future work.

6b. Did this change with respect to expression level of the VAP construct? If mitochondrial partners are the limiting factor in VAP localization to the ERMCS, then overexpression of exogenous VAP will saturate the endogenous binding partners, the bulk of the tracked VAP proteins will be less likely to interact, and the average diffusion coefficients will be artificially altered.

This is a great point. We have now explicitly quantified VAPB interactions as a function of expression level in Extended Data Fig. 3. As the reviewer suggests, the likelihood of VAPB binding at ERMCSs decreased weakly as the expression level of VAPB increases, particularly at the highest expression levels. However, this did not affect our analysis of ERMCS interactions, since molecules that do not show explicit ERMCS interaction behaviors (i.e., trajectory segments with sufficiently low diffusivity index) are not utilized to extract diffusion values. While freely diffusing VAPB molecules that transiently pass the contact site are still included in the neighborhood analysis, we caution against reading too deeply into the numbers extracted from that approach, since the drift term is left unconstrained and the resulting D_{eff} values are qualitative rather than quantitative (see Supplementary Text Section 6c). As such, a larger fraction of freely diffusing molecules within a single tessellation simply decreases the signal to noise for resolving substructure. Since we observe clear substructure anyway, we are not concerned about this source of error.

Minor Comments

- In Extended Data Figure 4, 'are indicated with black dots and blue arrows' should be changed to '...black dots and yellow arrows.'

We thank the reviewer for catching this error, it has been corrected.

- There was no literature list attached for the main text in the material I received. I received a literature list but only for the methods section and the expanded data section. I was able to find the missing literature section in the bioRxiv version of this paper, however.

We have corrected the mistake in the resubmission.

Referee #3:

The manuscript by Obara et al. visualized the transient, small and sensitive dynamic molecular bonds within mito-ER contact sites by using sptPALM. They also characterize ER-

mito contact site (ERMCS) morphology using FIB-SEM imaging. They show that the VAPB molecule, which is the main ER-resident tether, is dynamic but shows enhanced enrichment and reduced diffusivity in zones that correlate with ER-mito proximity. They use the VAPB enrichment to define the contact site and show that the size of the contact site was sensitive to the abundance of the mito tethering molecule PITPIP51 but not to the abundance of VAPB. They also show that contact sites expand under starvation conditions, but the VAPB molecules remain dynamic. Finally, they examine ALS mutants of VAPB and show that they have subtle changes in their contact site diffusivity, becoming trapped in subdomains.

Overall, the manuscript is a tour de force imaging achievement. The visualization of VAPB dynamics together with markers of ER and mitochondria is highly technically challenging and the authors carried out a lot of optimizations for the data acquisition to achieve this task. In addition, they did a careful job with the analysis of the sptPALM data, considering several complications such as the impact of the ER geometry on the molecular motion and using appropriate analysis algorithms to extract the dynamic parameters. The FIB-SEM images are also impressive. All that being said, the methodologies are not really new and it is not clear that a lot of new biological insight has been gained from the technical tour de force. The main conclusion is that the tethering molecules are dynamic rather than stationary. This is not too surprising given what we know about many other similar tethering or adhesion molecules. For example, molecular machinery within focal adhesions is highly dynamic, the synaptic adhesion molecules (neurexin and neuroligin) are highly dynamic. Hence, the model of maintaining an adhesive contact structure through dynamic interactions of adhesion molecules is not new. In fact, it would have been much more surprising if VAPB was stationary at the contact site. In addition, the part about the role of VAPB mutants in ALS is highly speculative. The authors show some subtle changes to the mobility of VAPB at contact sites but the mechanism for these changes and the physiological relevance to dysfunction in ALS are not explored. The work remains descriptive and for a high impact study, it would have been worth exploring how perturbing the tethering dynamics or the area of contact site impact ERMCS function and cell physiology.

We thank the reviewer for the positive remarks regarding our imaging approaches. We respectfully disagree, however, that our approaches and findings are not really new. Our data provide the first direct observations of the behavior and organization of tethering machinery comprising ERMCSs in living cells. These structures have been previously difficult to study because of the lack of non-perturbative techniques for detecting them in live cells. To overcome this, we integrated two different super-resolution imaging pipelines- SPT and FIB-SEM- to correlate structure and function at ERMCSs. This allowed us to extract information on molecular motion within the ER's complex topology that has never previously been achieved. In combination, these approaches paint a new picture of ERMCS biology— where these interfaces are stably maintained by the rapid exchange of individual tethers on millisecond time scales, leading to a metastable, nanoscale structure that can rapidly be adapted to changing cellular needs. Of note, the VAPB dynamics we report here are 5-50 times faster than seen for the neurexin and neuroligin examples cited by the reviewer (e.g., PMID: 26446217). Additionally, we show that despite this dynamism, ERMCSs maintain spatially organized subdomains in the steady state, enabling contact site remodeling in response to different physiological conditions. Underscoring the importance of this

metastability, we show disease-causing mutations in VAPB disrupt this architecture and plasticity. The unprecedented dynamic exchange of components and exquisitely maintained nanoscale structure that we report here present a new paradigm for understanding ERMCSs and will help direct future work aimed at characterizing the biology and function of contact sites.

Below are more specific comments for the authors:

Major comments:

1. I am a bit confused about how contact sites are identified. In the manuscript the authors comment that “we looked for locations where ER-localized tethers exhibited patterns of reduced single molecule motion, reasoning these may reveal the sites where they interact with their mitochondrial binding partners.” They then quantify the diffusivity of VAPB and show that it is reduced at contact sites. This seems like a circular logic if they are identifying contact sites as those where VAPB is less mobile. I may be confused about this, but it needs some clarification.

In our revision, we now more clearly clarify how contact sites are identified in our SPT data. Contact sites are initially identified and refined based on the probability mass function derived from VAPB trajectories, which makes no assumptions about the motion of the molecules. In the revised text, we have corrected our description of the methodology to be precise, and we have updated the methods section appropriately. We have also added a derivation of the probability mass function to the supplement, detailing how this was performed (Supplementary Text, Section 7b).

2. “These probability “hot spots” were present in the VAPB dataset but not with Halo-TA (Fig. 1f), and they were consistent with the size of ERMCSs extracted from FIB-SEM volumes (Fig. 1g-i).” This sentence is misleading because reading more deeply into the figure caption and SI, it seems that the authors measured contact site in FIB-SEM indirectly by simulating single molecule trajectories projected onto the FIB-SEM images and what they are comparing is the contact site size measured from these simulated traces, not directly from the FIB-SEM images. It would be informative to also include the projected size of the contact site directly measured from the FIB-SEM images so that the reader can have an idea about how these two methods compare. In addition, I suggest rewording this part of the text to be more transparent about the simulation.

To address the reviewer's concern, we have corrected the main text to explicitly state that the comparison in Fig. 1i is performed with a reduced resolution version of the FIB-SEM data to approximate sptPALM resolution, and we have added the full resolution data as requested to Extended Data Fig.1d.

3. The statistics are poorly reported throughout the manuscript. I could not find the number

of cells or the number of biological replicates or the type of statistical test performed for the different quantifications. I suggest the authors add this information in the figure captions. In addition, it seems the statistical analysis is done by considering each trace or contact site as an independent data point rather than comparing the means of biological replicates. Subtle differences will seem artificially statistically significant simply due the large number of data points when a trace or contact site is considered for the statistical analysis. I suggest the authors use super plots and report statistics on biological replicates.

We have now added all statistical results to the figure captions, as requested. Although we appreciate the suggestion of superplots, we find them to be more confusing than useful in data such as ours, which have many levels of statistical nesting (including different experiments, cells, contact sites, neighborhoods, trajectories, and binding events). In general, the primary source of variation in our data is between cells in the same dish and between contact sites in the same cell. This variation essentially covers the entire spectrum of potential states (see Extended Data Fig. 3). Indeed, as shown in the figure below for the reviewer, the variation between cells and between contact sites within single cells strongly dominates sample-to-sample variation in technical replicates. Thus, whenever possible we group the contact site data by cell for statistical purposes. We note that there are no statistically significant differences between experimental preparations in our data. Trajectories and neighborhoods are never used as independent data points. We have added a section to the supplementary methods describing the approach used for all figures throughout the paper (Supplementary Text, Section 7e).

Figure 1 for Reviewer 3. Methods of ERMCS diffusion analysis using effective 2D diffusion coefficients. a, The local diffusivity of VAPB molecules in a region of the peripheral ER that contains a single contact site with a mitochondrion (mitochondrion not shown). Fitted localizations within the local neighborhood are traced in magenta in the inset.

b, The reduction in local 2D D_{eff} across ERMCSs shows slight but significant decrease in effective motion within contact sites when data are appropriately paired so each ERMCS is compared to its neighborhood ($n=160$ contact sites, $p=0.0015$, Wilcoxon signed rank test).
c, The effective diffusion coefficient within ERMCSs and their local neighborhoods, color-coded by separate dishes independently transfected and imaged on the same day. Note the variation between cells and between contact sites within single cells (Extended Data Fig. 3) strongly dominates sample-to-sample variation in technical replicates.

4. Figure 4h-j, the observation that the P56S mutant explored only subdomain of the contact site should be supported by a quantification of area of confinement in comparison to the overall contact site area. This part of the manuscript is mainly supported by anecdotal examples. From the examples shown, it is not clear how the mutant is different from the WT and what exactly the reader should be seeing in the color-coded segmentations. Is there a hypothesis as to why the P56S only surveys subdomains if it is not aggregated in the contact site? The mechanism or the physiological relevance of this observation are unclear.

We thank the reviewer for these suggestions. In our revised manuscript, we now provide quantification of trapping of P56S VAPB molecules within ERMCSs (see Fig. 4f) and revise the discussion to clearly state their significance in light of the existing literature. This trapping causes a significant delay in the amount of time individual molecules spend at a single ERMCS, extending it beyond the resolution of our technique to quantify (Fig. 4g). It also causes significant increases in ERMCS size and decreases in the net diffusion landscape (Extended Data Fig. 9), as trapped molecules dominate the contact site landscape, precluding the normal dynamic exchange with the surrounding ER. As the reviewer suggests, one possible explanation is that small subsets of VAPB-P56S are aggregated at these sites. Another possibility is that VAPB-P56S has dimerized or oligomerized with VAPB at this site and consequently has enhanced tethering affinity with PTPIP51, no longer requiring multiple low affinity interactions. There is precedent in the literature for both of these explanations (PMIDs: 18713837, 20207736, 22131369, 28108526). We have adjusted the main text to expand on these possibilities and have added two sections in the supplement to fully discuss the literature in this area and the implications of our findings for the field (Supplementary Text, Sections 2d + 6c).

Minor comments:

5. Can author specify why distance between mitochondria and ER less than 24 nm was chosen as a cutoff for defining contact sites?

This number was chosen as an upper bound for the predicted tethering distance of VAPB-PTPIP51 based on our observations in PTPIP51-overexpressing cells and the results shown in PMID: 24893131. We have added a section to the supplement to detail this (Supplementary Text, Section 3b) and now explicitly state this in the main text.

6. "The affinity of VAPB for FFAT-containing binding partners like PTPIP51 is very low^{21,47,48}, suggesting contact site interaction may consist of very many rapid binding and unbinding events across the structure". The authors may consider extracting off rates to compare to the reported low affinity, and if such affinity (k_d) is known, k_{on} can be also estimated, which would provide a more quantitative view of VAPB dynamics on contact sites.

We do not have accurate numbers on the actual molecular density of VAPB or PIPTP51 at contact sites, so we cannot accurately calculate K_d or K_{on} values for the interaction of these two molecules. However, we now report the dwell time of VAPB in contact sites (median time ~ 500 msec) and the effective rate that molecules leave the contact site. This measurement does not directly report VAPB engagement to PTPIP51, since VAPB can also dimerize and may bind/unbind multiple times during its time in the ERMCS. This is indicated in the main text of the revised paper and added to Fig. 2.

Referee #4 (Remarks to the Author):

The manuscript presents data on the movement of the mitochondria-ER contact (ERMCS) tethering protein VAPB. Through single-molecule tracking, the authors show fascinating data that demonstrate that a fusion protein based on this tether occasionally shows targeted overlap of its signal with ERMCSs. This is not seen with a minimally tail-anchored protein. The strength of the manuscript lies in the single molecule tracking of VAPB that provides important and exciting insight about its extent of movement within and outside of ERMCSs. This makes Video 3 a true highlight of this study and membrane contact site (MCS) research in general.

Unfortunately, while I am impressed with this aspect of the study and also enthusiastic about any improvements that the authors will do, the study in its current form omits several aspects of what is known about ERMCS biology, for instance the distinction of rough and smooth ERMCSs or the role of autophagy for ERMCSs. This is reflected in the omission of key studies in the reference list, which also appears not matched to the text. The authors touch upon ERMCS biology by investigating starvation-relevance of their observation. However, this aspect is currently much underdeveloped, relies on a very long incubation period in starvation medium that is known to cause autophagy and does not address other, better characterized mechanisms that could give cleaner readouts. The study is also relying exclusively on one type of ERMCS tether, but many more are known, which reduces the overall relevance for the field. Another attempt to introduce significance into their findings uses a known ALS

mutation of VAPB. While this dataset is strong, it is not mechanistically explained and could suffer from the known role of this mutant to alter overall ER structure and lipid composition. Lastly, another major weakness of the study is that no knockouts or knockdowns of well-known factors for ERMCS formation have been put into context of the VAPB behavior.

Together, these deficiencies currently do not allow much mechanistic insight.

Overall, very strong descriptive data on VAPB are in the manuscript that is a candidate for nature. Unfortunately, the manuscript currently is underdeveloped, and contains no experiments that provide a minimum of information about the universality of the

observations. Once at least a few of such experiments are provided, the study would match the claims made in the title.

We thank the reviewer for their kind comments about the contributions of our sptPALM data to the contact site field, and we agree that the results are likely to be broadly impactful to scientists in diverse disciplines interested in interorganellar communication and regulation. We respectfully disagree, however, that the work is not of broader interest without characterizing every other putative tether and proposed ERMCS subtype. VAPB has a well-defined interaction with PTPIP51, is not perturbed by Halo-tagging, is free from interfering VAPB molecules in other membranes, and does not change contact site abundance when overexpressed. Other ERMCS tethers that we have so far examined lack these qualities, and so will require significant further work in order to use them in single particle tracking studies to probe ERMCS form and function. The work in this paper using VAPB as a prototypic tether provides the first clear example of the nanoscale structure and dynamics of ERMCSs. We demonstrated ERMCSs exhibit unprecedented dynamic exchange of components, have exquisitely maintained nanoscale structure, and show an incredible plasticity in adapting to cellular needs. Moving forward, these findings will provide a framework for others interested in contact sites, by opening up new approaches and concepts in this field. While the other questions the reviewer has suggested are of great interest and can be addressed in future work, the findings presented here are clearly of significant impact for the field in their own right.

Specific points

1. The field of ERMCSs has recently seen a critical refinement, where distinctions between rough ERMCSs and smooth ERMCSs have been distinguished (see Anastasia et al. Cell Reports 2021). Unfortunately, the authors do not provide any information about this property of the structures they describe in their initial characterization of ERMCSs. Accordingly, they describe their data as “this view of ERMCS ultrastructure”, but due to the aforementioned point, this is therefore an overstatement.

We thank the reviewer for this comment. We agree that rough-ERMCSs are a topic of great interest for future work, but they cannot be addressed by our current approach. Rough-ERMCSs are very rare in the regions we can image at the periphery of cells. In fact, we barely observe any at all at the ER periphery in our FIB-SEM data. One possible explanation for this is that expanded rough-ERMCSs as described by Anastasia et al. are enriched in specialized cell types as most of the literature describing these structures focused on specific tissues, like the liver. However, we agree with the reviewer that the distinction between rough- and smooth-ERMCSs is important, so we have updated the text to more clearly detail the reasoning for selecting the smooth ERMCSs that are dominant in our system and have added a section to the supplementary materials discussing the literature in this area (see Supplementary Text, Section 2a). As suggested by the reviewer, we revise the sentence “this view of ERMCS ultrastructure” in the third paragraph to avoid any initial confusion about the complexity of ERMCSs.

2. While I have no objection to the use of VAPB, since this is an important ERMCS tether, the manuscript falls short in terms of an investigation of its relative importance. The authors initially state that it “likely” acts as “a tether for other organelles”. This is a rather casual observation and decreases the otherwise quantitative aspects of this study. At a minimum, the authors should give an idea about the frequency of ERMCS localization of VAPB versus other enrichments early on. This is provided in part by Figure 3F. However, this figure lacks a calculation of the relative abundance of ERMCS-localization of VAPB versus other contact sites.

We appreciate the reviewer’s concern, and have now added Extended Data Fig. 2, which fully quantifies the frequency and VAPB-binding propensity of ERMCSs vs other contact sites.

2b. Also, I am not sure about the significance of the presented data. While it is true that in the presence of high amounts of PTPIP51, the number of non-mito MCS decreases, it is unclear whether the data presented by the authors actually show a specific reduction of the relative amounts of VAPB on non-mitochondrial contact sites. Could the authors produce a “coefficient” that measures relative amounts on both types of MCS?

As requested, we have developed probabilistic coefficients that measure the relative amounts of VAPB at each type of contact site, and a collective “mitochondria enrichment coefficient” that directly quantifies propensity of VAPB to bind at ERMCSs versus other sites. These are now utilized in Fig. 3 and Extended Data Figs. 3, 8, +9. The detailed derivation and methodology is now given in Supplementary Text, Sections 7b-d. As expected, PTPIP51 overexpression causes a dramatic redistribution of VAPB from other contact sites to mitochondria.

2c. Moreover, it would be better if they actually identify the nature of those enrichments that do not contact mitochondria.

Extensive literature has already demonstrated that VAPB has interactions with many other compartments, and the specific binding partners on each are known (reviewed in Murphy et al., 2016, PMID: 26898182). These interactions with other organelles are believed to be less prevalent than VAPB-mediated interactions with mitochondria, an observation our findings also support (Extended Data Fig. 2). We agree it would in principle be very interesting to know the abundance and prevalence of these different types of ER contact sites in single cells, however, we feel this falls outside the scope of this work as we are not aware of any existing technology that would support both high speed single molecule imaging and the 5+ colors that would be needed to label each of VAPB’s potential compartments of interaction.

3. The extent of ER membrane curvature is correlated to distinct lipid species. For instance, PE could mediate increased curvature and microscopic techniques exist to localize membrane tension developed by the Matile and Roux labs (e.g., Colom et al., Nature Chemistry 2018).

This is an interesting idea, but this kind of multiplexed experiment is well beyond the capacity of existing microscopy technologies. The curvature the reviewer mentions occurs over spatial scales of a few nanometers, significantly beneath the resolution of any imaging approach that could use the proposed tension sensors. The probes mentioned are lifetime-based sensors that require a scanning point implementation using pulsed lasers, and as such are neither compatible with sptPALM nor the other superresolution techniques we have access to (even depletion technologies perturb lifetimes in non-linear ways, making calibration very challenging, if not impossible). Additionally, since the contact sites themselves are generally subdiffraction-limited in at least one dimension, use of these probes in such a confined space will also, by necessity, dilute the signal with the surrounding ER or mitochondria (and probably the back side of the same organelle), making reliable readings impossible. Should technology like this become available for use in an electron microscopy-based approach, we would be very keen to pursue this in future work.

4. Many physiological conditions exist, which increase ERMCSs. A key mechanism identified by the Hajnoczky and Lavandero labs is short-term ER stress. I am surprised that the authors do not investigate this mechanism. Similarly, the interference with curvature-inducing PE using, for instance cinnamycin or a PSD1 mutation could yield more relevant findings.

We agree that examining ERMCSs under various other physiological conditions beyond what we have done would be interesting, and we plan to do this in the future. However, all of the conditions suggested by the reviewer are known to have significant effects on underlying ER morphology that would need rigorous controls. We would need to do FIB-SEM on each of the different conditions to clarify changes in ER and ERMCS ultrastructure, and we would have to validate our trajectory analysis pipeline accordingly. Given that the goal of our paper is to illustrate the use of FIB-SEM and sptPALM analysis correlated with organelle imaging for understanding the relationship between tether dynamics and contact site structure, we feel that the examples we have provided for perturbing ERMCS structure should be sufficient.

4b. Instead, they focus on nutrient deprivation. This condition has the disadvantage of also leading to autophagy, for instance the authors should have a look at the study by Hamasaki et al., nature 2013. This choice therefore requires investigation of autophagosome localization relative to ERMCSs and their amounts. Another issue is that during a time course of starvation, more or less ERMCSs could appear due to this issue. A control of the extent of ERMCS formation during the very long 8h timeperiod chosen by the authors should

control for this possibility. Moreover, they again do not consider the data recently reported that distinguish between rough and smooth ERMCSs.

While we recognize that basal, steady-state autophagy levels are elevated in conditions of nutrient deprivation, we are not aware of any evidence that VAPB is involved in regulating these autophagic events. We have recently performed FIB-SEM on cells that underwent this same nutrient deprivation treatment at 8 hours (in contrast to the Hamasaki study, which is performed at 2 hours), and we did not observe any autophagosomes at ERMCSs under these conditions, rather, they were largely at ER exit sites (Liao et al., see preprint at: https://papers.ssrn.com/sol3/papers.cfm?abstract_id=4144963). This agrees with a significant body of prior work in Yeast, showing starvation-induced autophagosomes are associated with early secretory components, not ER-mitochondria contact sites (PMIDs: 23904270, 23930225, 30787039). To alleviate the reviewer's concern that additional types of ERMCSs could form in starved cells and confound our data, we quantified any change in number of ERMCSs under our starvation conditions, which is now included in Extended Data Fig. 8. There was no significant change observed under these conditions.

5. While the data on the VAPB mutant are strong, they are not mechanistically explained. How does this mutant achieve this change in motility? It is likely misfolded, so are we just looking at the consequence of this? More questions arise when considering that this mutant disrupts normal ER morphology, thus suggesting that the observations are a side-effect of this property.

We thank the reviewer for this suggestion. We have updated the discussion and added a section to the supplement (Supplementary Text, Section 2d) to fully discuss the potential basis of this observation.

6. Overall, the decision to exclusively focus on the VAPB tether limit the interest in this study right now. While many presented datasets are very strong, they are restricted to this tether and do not provide general information about tethers or ERMCSs. This limitation is puzzling, given that many more tethering proteins are known, in addition to many other factors that are required for ERMCS formation, and at least a couple should be analyzed in a similar manner to give readers information that is not restricted to VAPB. This is particularly problematic, given the title promises much more than just a study about VAPB. At an absolute minimum, the authors must be more honest with the title and clearly declare that the manuscript deals with one tether type, the VAPB-PTPIP51 pair. Such a decision would, however, reduce the enthusiasm that I otherwise have for the experiments presented in manuscript.

While we appreciate that the reviewer is also excited to see how other ERMCS tethers behave and are regulated at this interface, single molecule tracking data can be easily misinterpreted or erroneously constructed from localizations if the experimenters are not

able to effectively create negative and positive controls. VAPB is amenable to this because its interaction with PTPIP51 is well-characterized, not perturbed by fluorescent tagging, not dependent on VAPB expression levels, and free from interfering VAPB molecules in other membranes. Unfortunately, these same characteristics are not true for other known ERMCS tethers. While we agree that these results will be very interesting, we do not feel they can be reasonably achieved within the constraints of this paper and would require many additional years of work. We believe our findings are highly worthwhile even without examining other tethers beside VAPB. The remarkable dynamics of VAPB that we report are nearly an order of magnitude greater than that reported in all prior work examining the dynamics of ER-PM tethers and adhesion factors. This puts an entirely new perspective on how ERMCSs are formed, shaped, and maintained. ERMCS structures must now be viewed as highly dynamic steady-state systems in which subtle changes in tether abundance can lead to rapid and dramatic effects on contact site organization. We demonstrate these effects using conditions known to alter ERMCSs, including starvation and overexpression of PTPIP51. Further insight into ERMCS biology relevant to human disease is provided by examining the effects of a P56S pathogenic mutation of VAPB, which disrupted the normal diffusion landscape of VAPB molecules across the contact site, leading to molecules unable to leave it. There is clearly much more we and others can now do with this powerful system in dissecting ERMCSs as well as other contact site systems within the cell, but what we have introduced in this paper is clearly impactful to the field. We have altered the title of the paper as suggested by the reviewer.

7. Previous studies on ERMCSs in nature have used knockouts or knockdowns for key ERMCS regulatory proteins (e.g., de Brito et al. 2008, Hamasaki et al., 2013). The current study contains none, thus seriously limiting the scope of the study.

We appreciate this comment, but for the reasons outlined above, we feel this analysis is best pursued in future work. The perturbations performed by the Scorrano and Yoshimori labs in these studies are likely affecting other targets besides VAPB at ERMCSs, so interpretation of the results would be complicated without the ability to also track other factors affected by these perturbations, which we believe is beyond the scope of this study.

Minor points:

1. The authors chose to use a highly uncommon abbreviation for mitochondria-ER contacts. The field has homed in on the terms MAMs for biochemical isolates of these contacts and MERCs for the physical contact sites. The vast majority of the existing literature uses these terms. It is not a good idea to keep changing terms for this structure and the authors must use the consensus abbreviation, especially in an important journal such as nature.

We thank the reviewer for the suggestion, but we feel this is only true for the mitochondria-focused part of the literature. The ER contact site and ER-localized tether field outside of mitochondria has extensively utilized a generalized naming system with the ER listed first since the primary tether being examined is localized there (e.g., ER-PM junctions, ER-

vacuole junctions, ER-lysosome contact sites, etc.). As our data is focused on the ER side of the interaction and VAPB behavior in the ER membrane, we feel it important to retain the term ER-mitochondria contact sites (ERMCS).

2. It looks like the reference numbering is wrong. For instance, the authors refer to #21 and #44 for the ALS mutants of VAPB but those references do not deal with this question. In fact, from the list provided by the authors, I am unable to identify the papers where this had been identified.

We thank the reviewer for noticing this and have corrected the referencing.

Reviewer Reports on the First Revision:

Referees' comments:

Referee #1 (Remarks to the Author):

This is a resubmission so I am going to only address the author's revision and rebuttal comments. My own comments and concerns were very thoroughly addressed, and in general the answers to all reviewer's comments were astonishingly detailed and thoughtful. It is no wonder that this revision took some time.

I believe that this huge work, with the combination of single-particle tracking and FIB-SEM volume imaging to correlate structure and function, is really novel and exciting. In addition, the careful and highly detailed descriptions of the methodology, and especially the discussion of pitfalls in the supplementary material, make this manuscript an important contribution that will have serious impact for other researchers investigating organelle-organelle contacts.

I recommend publication without further revision.

Referee #2 (Remarks to the Author):

The authors have addressed my comments

Referee #3 (Remarks to the Author):

The authors have addressed all my technical comments with the revisions. The novelty over previous work is also more clearly delineated in the revised manuscript.

Referee #4 (Remarks to the Author):

Obara et al have revised their manuscript. The authors have now stated more clearly that they focus on VAPB, already in the title. This has reduced the scope of their study but has now tightened the story. I agree that a wholesome examination of MERCs would be too much for a Nature manuscript, the issue was simply that the authors previously claimed they do so. The authors make a very compelling case in the response to my comments and also in the manuscript itself that this remains a fully Nature-worthy story. I agree with this assessment. In fact, the story is now in a much better shape and I am sure the readers will find it extremely insightful. As mentioned by the authors, it will no doubt lead to multiple follow-ups that I expect to be equally exciting. To sum it up, I believe the authors have done an outstanding job of addressing all of my concerns, as well as the concerns of the other reviewers. The manuscript in its current form will be a very valuable tool for future

studies!

Only very minor comments remain. The supplementary text contains many valuable statements about the methods used to study this and other types of MCS that could enter the main text.

Major points:

1. It would be useful to move a condensed version into the main text, a few sentences, containing the main statements on FRET sensors, and MERC subclasses. This would put the study into context and would give the important statements made in the supplementary text the necessary exposure.
2. How does the Halo-tag approach compare to split GFP approaches? This should be spelled out more clearly.
3. The characteristics of the P56S mutant should be more clearly spelled out in the concluding statement of the Results section.
4. While I am ok with the authors using ERMCS as an abbreviation, my main concern regarding the use of acronyms is for the benefit of the reader. I recommend adding terms like MAMs and MERCs to the introduction, with the intent of clarifying potential differences in their usage and of facilitating to find this essential resource, especially for readers unfamiliar with the topic.

Minor points:

1. The font size in some of the figures is really small, making reading them difficult.
2. Extended data figure 3: Likelihood is mis-spelled. What is the arrowhead pointing to in extended data figure 3c?

Author Rebuttals to First Revision:

Referees' comments:

Referee #1 (Remarks to the Author):

This is a resubmission so I am going to only address the author's revision and rebuttal comments.

My own comments and concerns were very thoroughly addressed, and in general the answers to all reviewer's comments were astonishingly detailed and thoughtful. It is no wonder that this revision took some time.

I believe that this huge work, with the combination of single-particle tracking and FIB-SEM volume imaging to correlate structure and function, is really novel and exciting. In addition, the careful and highly detailed descriptions of the methodology, and especially the discussion of pitfalls in the supplementary material, make this manuscript an important contribution that will have serious impact for other researchers investigating organelle-organelle contacts.

I recommend publication without further revision.

We thank the reviewer for all the wonderful suggestions throughout the process.

Referee #2 (Remarks to the Author):

The authors have addressed my comments

We thank the reviewer and appreciate all the guidance in bringing the manuscript to its current form.

Referee #3 (Remarks to the Author):

The authors have addressed all my technical comments with the revisions. The novelty over previous work is also more clearly delineated in the revised manuscript.

We thank the reviewer for all of the suggestions, particularly the guidance on recognizing the importance of the high speed dynamics.

Referee #4 (Remarks to the Author):

Obara et al have revised their manuscript. The authors have now stated more clearly that they focus on VAPB, already in the title. This has reduced the scope of their study but has now tightened the story. I agree that a wholesome examination of MERCs would be too much for a Nature manuscript, the issue was simply that the authors previously claimed they do so. The authors make a very compelling case in the response to my comments and also in the

manuscript itself that this remains a fully Nature-worthy story. I agree with this assessment. In fact, the story is now in a much better shape and I am sure the readers will find it extremely insightful. As mentioned by the authors, it will no doubt lead to multiple follow-ups that I expect to be equally exciting. To sum it up, I believe the authors have done an outstanding job of addressing all of my concerns, as well as the concerns of the other reviewers. The manuscript in its current form will be a very valuable tool for future studies!

We thank the reviewer for all of the suggestions—they were very helpful in helping us identify the correct way to frame the work and putting it in context of the literature of the field.

Only very minor comments remain. The supplementary text contains many valuable statements about the methods used to study this and other types of MCS that could enter the main text.

Major points:

1. It would be useful to move a condensed version into the main text, a few sentences, containing the main statements on FRET sensors, and MERC subclasses. This would put the study into context and would give the important statements made in the supplementary text the necessary exposure.

We thank the reviewer for the suggestion. We have now updated the introduction to explicitly state these things, and added some references to assist with clarity.

2. How does the Halo-tag approach compare to split GFP approaches? This should be spelled out more clearly.

We have updated Supplementary Information, Section Id to incorporate the split GFP approaches and clarify this difference.

3. The characteristics of the P56S mutant should be more clearly spelled out in the concluding statement of the Results section.

We appreciate the suggestion, we have adjusted the text to clarify this point.

4. While I am ok with the authors using ERMCS as an abbreviation, my main concern regarding the use of acronyms is for the benefit of the reader. I recommend adding terms like MAMs and MERCs to the introduction, with the intent of clarifying potential differences in their usage and of facilitating to find this essential resource, especially for readers unfamiliar with the topic.

We thank the reviewer for this suggestion. The main text has been updated to mention the common other term (MERCs) and we have added Supplemental Information, Section 2A to clearly distinguish these from MAMs, with appropriate explanation.

Minor points:

1. The font size in some of the figures is really small, making reading them difficult.

We thank the reviewer for catching this, it has been corrected.

2. Extended data figure 3: Likelihood is mis-spelled. What is the arrowhead pointing to in extended data figure 3c?

We thank the reviewer for catching this, it has been corrected, and the legend of Ext. Data Fig 3 has been amended to explain that this referred to the single cell example in Ext. Data Fig. 3d.